# Can Microcanonical Langevin Dynamics Leverage Mini-Batch Gradient Noise?

Emanuel Sommer [* 1 2]   Kangning Diao [* 3 4]   Jakob Robnik [3]   Uroš Seljak [3 5]   David Rügamer [1 2]

## Abstract

Scaling inference methods such as Markov chain Monte Carlo to high-dimensional models remains a central challenge in Bayesian deep learning. A promising recent proposal, microcanonical Langevin Monte Carlo, has shown state-of-the-art performance across a wide range of problems. However, its reliance on full-dataset gradients makes it prohibitively expensive for large-scale problems. This paper addresses a fundamental question: Can microcanonical dynamics effectively leverage mini-batch gradient noise? We provide the first systematic study of this problem, establishing a novel continuous-time theoretical analysis of stochastic-gradient microcanonical dynamics. We reveal two critical failure modes: a theoretically derived bias due to anisotropic gradient noise and numerical instabilities in complex high-dimensional posteriors. To tackle these issues, we propose a principled gradient noise preconditioning scheme shown to significantly reduce this bias and develop a novel, energy-variance-based adaptive tuner that automates step size selection and dynamically informs numerical guardrails. The resulting algorithm is a robust and scalable microcanonical Monte Carlo sampler that achieves state-of-the-art performance on challenging high-dimensional inference tasks like Bayesian neural networks. Combined with recent ensemble techniques, our work unlocks a new class of stochastic microcanonical Langevin ensemble (SMILE) samplers for large-scale Bayesian inference.

---

*Equal contribution [1]Department of Statistics, LMU Munich, Munich, Germany [2]Munich Center for Machine Learning, Munich, Germany [3]Department of Physics, University of California, Berkeley, USA [4]Department of Astronomy, Tsinghua University, Beijing, China [5]Physics Division, Lawrence Berkeley National Lab, Berkeley, USA. Correspondence to: David Rügamer <david@stat.uni-muenchen.de>.

*Proceedings of the 43rd International Conference on Machine Learning*, Seoul, South Korea. PMLR 306, 2026. Copyright 2026 by the author(s).

## 1. Introduction

The quest for more efficient and robust Markov chain Monte Carlo (MCMC) samplers is central to advancing high-dimensional Bayesian inference. For years, Hamiltonian Monte Carlo (HMC; Duane et al., 1987; Neal, 2011) has been the dominant paradigm for navigating the complex posteriors of modern machine learning models. However, the recently proposed microcanonical HMC (Robnik et al., 2023) and its Langevin-based counterpart, the microcanonical Langevin Monte Carlo (MCLMC; Robnik & Seljak, 2024) sampler, have proven to be a powerful new alternative. By simulating dynamics on a constant-energy surface, MCLMC is uniquely equipped to explore challenging posteriors, such as those of a Bayesian neural network (BNN), much faster than traditional HMC-based methods (Robnik et al., 2024; Sommer et al., 2025).

Despite its promising results, MCLMC, just like most MCMC algorithms, faces a critical limitation that has so far confined its impact and application: its reliance on full-dataset gradients. This makes it computationally infeasible for large-scale problems omnipresent in modern machine learning. Despite the existence of other stochastic gradient MCMC (SGMCMC) methods (Welling & Teh, 2011; Chen et al., 2014; Girolami & Calderhead, 2011), dealing with the gradient noise induced by mini-batching the dataset, MCLMC behaves differently in noisy systems. Robnik & Seljak (2024) showed that Langevin noise in MCLMC does not require a corresponding damping term for convergence and does not affect the stationary distribution. As the noise in stochastic gradients is often assumed to resemble Langevin noise, this suggests that mini-batched MCLMC may work *without* any additional noise injection. This stands in contrast to stochastic HMC or Langevin samplers, which in practice are typically run in a heavily over-damped regime, likely at the cost of efficiency. Rather, our perspective is closer to Langevin-type analyses of stochastic optimization dynamics (Mandt et al., 2017), while considering fundamentally different dynamics. This research gap motivates the central question of our work:

*Can microcanonical dynamics leverage mini-batch gradient noise, and thereby be made scalable to modern deep learning?*

This paper presents the **first systematic study of stochastic**

**microcanonical Langevin dynamics**, particularly focused on BNNs, and makes several contributions listed below.

**Our Contributions**

- We first demonstrate that naive adaptations of MCLMC with stochastic gradients are insufficient.

- We establish a rigorous theoretical foundation for microcanonical dynamics under stochasticity; specifically, we formally derive the systematic bias induced by anisotropic gradient noise and prove that a principled preconditioning scheme eliminates the resulting noise-induced drift.

- Algorithmically, we address numerical instability of minibatch MCLMC in high dimensions by introducing a novel energy-variance-based adaptive tuner that ensures robust performance and reduces hyperparameter sensitivity.

- Taking these findings together, we propose a scalable and effective mini-batch version of MCLMC.

- Finally, our empirical evaluation across a diverse set of BNN applications validates that our method is robust and well-working in high dimensions, yielding state-of-the-art performance.

## 2. Background & Related Work

We consider the general problem of Bayesian inference for high-dimensional parametric models such as BNNs. The goal is to infer the posterior distribution over the model's parameters $\boldsymbol{\theta} \in \Theta \subseteq \mathbb{R}^d$. Given a prior density $p(\boldsymbol{\theta})$, and observed data $\mathcal{D} = \{(\boldsymbol{x}_i, \boldsymbol{y}_i)\}_{i=1}^N \in (\mathcal{X} \times \mathcal{Y})^n$, the posterior density $p(\boldsymbol{\theta}|\mathcal{D})$ is given by Bayes' rule: $p(\boldsymbol{\theta}|\mathcal{D}) = p(\mathcal{D}|\boldsymbol{\theta})p(\boldsymbol{\theta})/p(\mathcal{D})$. Using the posterior predictive density (PPD), we can quantify the uncertainty of a prediction $\boldsymbol{y}^* \in \mathcal{Y}$ for a new data point $\boldsymbol{x}^* \in \mathcal{X}$ by integrating over the posterior distribution of the parameters $p(\boldsymbol{y}^*|\boldsymbol{x}^*, \mathcal{D}) = \int_{\Theta} p(\boldsymbol{y}^*|\boldsymbol{x}^*, \boldsymbol{\theta})p(\boldsymbol{\theta}|\mathcal{D}) \, d\boldsymbol{\theta}$. This integral is analytically intractable for most models, necessitating approximation methods.

### 2.1. Monte Carlo Sampling

Monte Carlo sampling provides a practical way to compute the posterior and PPD by numerically approximating its integral via samples from $p(\boldsymbol{\theta}|\mathcal{D})$. Given a set of $S \cdot K$ MCMC samples $\{\boldsymbol{\theta}^{(k,s)}, k \in [K], s \in [S]\}$ from $K$ independent chains, the PPD is approximated by its empirical counterpart $p(\boldsymbol{y}^*|\boldsymbol{x}^*, \mathcal{D}) \approx \frac{1}{K \cdot S} \sum_{k=1}^{K} \sum_{s=1}^{S} p\left(\boldsymbol{y}^*|\boldsymbol{x}^*, \boldsymbol{\theta}^{(k,s)}\right)$.

**Full-batch Sampling** Full-batch MCMC methods, such as HMC and its tuned variant, the No-U-Turn Sampler (NUTS; Hoffman & Gelman, 2014), are considered the gold standard for high-dimensional sampling (Štrumbelj et al.,

2024). They leverage gradients of the full-data likelihood to explore the posterior efficiently. However, these methods are computationally expensive as each step requires calculating gradients over the entire dataset. This makes them impractical for large-scale datasets. Furthermore, a significant challenge even for powerful HMC-based samplers is the difficulty of navigating complex and often highly multimodal loss landscapes of neural networks. While ensembles of many short and warm-started chains have been shown to improve exploration and efficiency (see, e.g., Sommer et al., 2024; 2025), they still rely on full-batch gradients.

**MCLMC** Recently, MCLMC has emerged as a state-of-the-art full-batch sampler, outperforming alternatives like the popular Hamiltonian Monte Carlo-based NUTS in analytical benchmarks (Robnik et al., 2024; Robnik & Seljak, 2024; Robnik et al., 2025), cosmological inference (Simon-Onfroy et al., 2025), and BNN inference (Sommer et al., 2025). MCLMC chooses a specific Hamiltonian $H(\boldsymbol{\theta}, \boldsymbol{\Pi})$ such that marginalizing the momentum $\boldsymbol{\Pi} \in \mathbb{R}^d$ at a fixed total energy yields the desired stationary distribution of $\boldsymbol{\theta} \in \mathbb{R}^d$. This dynamic is described by the following stochastic differential equation (SDE):

$$
\begin{aligned}
d\boldsymbol{\theta} &= \boldsymbol{u} \, dt, \\
d\boldsymbol{u} &= (1 - \boldsymbol{u}\boldsymbol{u}^\top)\big((d-1)^{-1} \nabla \log p(\boldsymbol{\theta}|\mathcal{D}) \, dt + \eta \, d\boldsymbol{W}\big),
\end{aligned}
\tag{1}
$$

where $\boldsymbol{u} = \boldsymbol{\Pi}/||\boldsymbol{\Pi}||$ is the momentum direction, $\boldsymbol{W}$ is the standard Wiener process, and $\eta$ is a free parameter that determines the distance traveled before momentum decoherence.

In practice, this SDE is solved using numerical integrators such as the Velocity Verlet algorithm (Leimkuhler & Matthews, 2015), which introduces numerical errors with each step. Since the ideal MCLMC dynamic conserves total energy $E$, the change in total energy per step

$$
\Delta E = \Delta \log p(\boldsymbol{\theta}|\mathcal{D}) - (d-1) \log(\cosh \delta + \boldsymbol{e}^\top \boldsymbol{u} \sinh \delta),
\tag{2}
$$

serves as a useful proxy for this numerical error, where $\boldsymbol{e} = -\nabla_{\boldsymbol{\theta}} \log p(\boldsymbol{\theta}|\mathcal{D})/||\nabla_{\boldsymbol{\theta}} \log p(\boldsymbol{\theta}|\mathcal{D})||$, $\delta = \Delta t ||\nabla_{\boldsymbol{\theta}} \log p(\boldsymbol{\theta}|\mathcal{D})||/(d-1)$, and $\Delta t$ is the integration step size. If the stepsize is too large, this error can destabilize the integrator and bias the samples. It is thus important to control the stepsize such that this bias remains sufficiently small compared to the variance arising from finite sampling, stochastic gradient bias, and the mixing efficiency of the chains. As discussed in Robnik et al. (2024), the energy error can be used to control the numerical error relative to these other sources.

### 2.2. Mini-batch Sampling

To overcome the scalability limitations of full-batch methods, SGMCMC algorithms were developed. These methods

use gradients computed on mini-batches of data, offering a significant speedup, similar to stochastic gradient descent in standard optimization of neural networks. Prominent examples include stochastic gradient Langevin dynamics (SGLD; Welling & Teh, 2011) and stochastic gradient Hamiltonian Monte Carlo (SGHMC; Chen et al., 2014).

Among the most effective extensions is scale-adapted SGHMC (Springenberg et al., 2016), which is widely recognized as a state-of-the-art SGMCMC baseline (see, e.g., Shi et al., 2025; Andrade & Sato, 2025, for recent studies). Scale-adapted SGHMC incorporates diagonal preconditioning (Roberts & Rosenthal, 2001; Haario et al., 2001), similar to the mechanism in the RMSprop optimizer, to automatically adjust the step size for each parameter. This adaptation is motivated by the complex geometry of a neural network's loss landscape, significantly enhancing both convergence and the ability to explore the posterior distribution. Due to its robust performance and efficient scaling, we select scale-adapted SGHMC as our primary baseline method. Unless otherwise specified, all references to SGHMC in our experiments refer to this enhanced version. Further discussion on adaptive SGHMC methods can be found in Section F.1.

Mini-batch samplers offer a more scalable approach, but their theoretical guarantees and sampling efficiency often differ considerably from their full-batch counterparts. A possible stochastic version of MCLMC faces several challenges, many distinct from those in traditional SGMCMC.

### 2.3. Improving Sampling for Neural Networks

One particularly challenging application for sampling-based inference are BNNs. Due to their complex and often highly multimodal loss surface, traversing the posterior with a sampler is challenging. To mitigate these problems, recent approaches propose to use optimized solutions as warm-starts for sampling (instead of, e.g., sampling from the chosen prior), lifting the sampling into regions of higher probability (Paulin et al., 2025). To tackle the multimodality of the posterior, ensembling methods using multiple chains from different starting locations have proven effective. Such approaches have been proposed both for HMC and for MCLMC (Duffield et al., 2025; Sommer et al., 2025). These methods, also called Bayesian deep ensembles (BDE), have shown state-of-the-art performance for BNN uncertainty quantification. We will thus not only study stochastic variants of MLCMC, but also microcanonical Langevin ensembles (MILE).

## 3. Stochastic Microcanonical Langevin Dynamics

In order to develop and study stochastic MCLMC and its ensemble variant MILE, we begin by analyzing its pitfalls

and derive the necessary remedies to establish it as a practical and scalable sampler for modern applications. In the following, we will therefore propose different variants of stochastic MILE (short: SMILE), with its most basic version denoted as SMILE-naive. Our later proposed extensions pSMILE-naive and pSMILE build on this basic version.

### 3.1. Sampling Without Explicit Noise Injection

To develop stochastic MCLMC, Equation (1) needs to be computed using mini-batches. This will introduce an error in the gradient estimation, differing from the averaged full-batch gradient. For a random mini-batch $\mathcal{B} \subseteq \mathcal{D}$ of the data $\mathcal{D}$ with $|\mathcal{B}| =: B < N$, assuming the gradient difference for sample $\mathcal{D}_i$ is $\boldsymbol{\varepsilon}_{\mathcal{D}_i} = \nabla_{\boldsymbol{\theta}} \log p(\boldsymbol{\theta}|\mathcal{D})/N - \nabla_{\boldsymbol{\theta}} \log p(\boldsymbol{\theta}|\mathcal{D}_i)$, the mini-batch gradient at fixed $\boldsymbol{\theta}$ is the sum of the single sample gradients,

$$\nabla_{\boldsymbol{\theta}} \log p(\boldsymbol{\theta}|\mathcal{B} \subset \mathcal{D}) = {}^B\!/\!_N \nabla_{\boldsymbol{\theta}} \log p(\boldsymbol{\theta}|\mathcal{D}) + \sum_{i \in \mathcal{B}} \boldsymbol{\varepsilon}_{\mathcal{D}_i}.$$

A typical assumption for the gradient noise (see, e.g., Ma et al., 2015; Zhu et al., 2019; Ziyin et al., 2022) is $\sum_{i \in \mathcal{B}} \boldsymbol{\varepsilon}_{\mathcal{D}_i} \sim \mathcal{N}(0, \mathbf{V}(\boldsymbol{\theta}))$, with the covariance $\mathbf{V}$ having an unknown dependence on $\boldsymbol{\theta}$.

A naive stochastic version of MCLMC (later in its ensemble variant referred to as **SMILE-naive**) can be defined as

$$\mathrm{d}\boldsymbol{\theta} = \boldsymbol{u}\,\mathrm{d}t,$$
$$\mathrm{d}\boldsymbol{u} = {}^N\!/\!_B(1 - \boldsymbol{u}\boldsymbol{u}^\top)(d-1)^{-1}\nabla_{\boldsymbol{\theta}} \log p(\boldsymbol{\theta}|\mathcal{B})\,\mathrm{d}t. \quad (3)$$

Here we omit the explicit noise injection because the noise already exists in $\nabla \log p(\boldsymbol{\theta}|\mathcal{B})$, which is typically rather large for a small batch size. Thus, additional noise is likely to slow down the convergence. In practice, the second-order Minimal-Norm integrator (Omelyan et al., 2002; 2003) is used to numerically solve (3) for all experiments in this work (further details in Section F.3).

> **Preliminary finding:** Stochastic microcanonical Langevin dynamics can be implemented with implicit mini-batch gradient noise injection instead of explicit noise injection.

### 3.2. The Pitfall of Anisotropic Stochastic Gradient Noise

Robnik & Seljak (2024) show that in continuous time, the stationary distribution of MCLMC is the target $p(\boldsymbol{\theta}|\mathcal{D})$ for any amplitude of injected isotropic noise. In Appendix A we extend this result to the stochastic gradient setting:

**Proposition 1** (informal). *If the mini-batching noise can, in continuous time, be modeled as an isotropic Wiener process, the theoretical properties of MCLMC (stationarity and geometric ergodicity) carry over to the continuous-time limit of a stochastic MCLMC sampler.*

However, mini-batch noise often has a position-dependent

*Table 1.* Bias $b^2(\downarrow)$ of different SGMCMC samplers on three 10-$d$ analytical posteriors with explicitly injected Gaussian noise. The reported bias is the average over 10 independent chains, and the standard deviation is the standard deviation of the average value obtained with bootstrap. The *Baseline* is full-batch MCLMC performance with optimal noise parameter $\eta$. Further details can be found in Section C.

| Target | Noise Type | SMILE-naive | SGLD | SGHMC | pSMILE-naive |
|---|---|---|---|---|---|
| **ICG** (Baseline: *0.0001*) | Isotropic | **0.003** $\pm$ 0.001 | 0.033$\pm$0.006 | 0.095$\pm$0.033 | 0.006 $\pm$ 0.001 |
| | Diagonal | 0.245$\pm$0.027 | 0.184$\pm$0.021 | 0.186 $\pm$0.020 | **0.038** $\pm$ 0.008 |
| | Correlated | 0.502$\pm$0.194 | 0.189 $\pm$ 0.029 | 0.235$\pm$0.067 | **0.055** $\pm$ 0.007 |
| | Spatially-varied | 0.157$\pm$0.010 | 0.328$\pm$0.011 | 0.331$\pm$0.011 | **0.093** $\pm$ 0.019 |
| **Rosenbrock** (Baseline: *0.0003*) | Isotropic | **0.002** $\pm$ 0.001 | 0.005 $\pm$ 0.001 | 0.004 $\pm$ 0.002 | 0.004$\pm$0.001 |
| | Diagonal | 0.302 $\pm$ 0.111 | 0.085 $\pm$ 0.007 | 0.160 $\pm$ 0.027 | **0.046** $\pm$ 0.002 |
| | Correlated | 0.265 $\pm$ 0.042 | 0.074 $\pm$ 0.007 | 0.085 $\pm$ 0.014 | **0.070** $\pm$ 0.005 |
| | Spatially-varied | **0.048** $\pm$ 0.005 | 0.079$\pm$0.013 | 0.095$\pm$0.013 | 0.052 $\pm$ 0.005 |
| **Funnel** (Baseline: *0.004*) | Isotropic | **0.014** $\pm$ 0.005 | 0.141 $\pm$ 0.019 | 0.128 $\pm$ 0.019 | 0.021 $\pm$ 0.005 |
| | Diagonal | 0.283 $\pm$ 0.146 | 0.063 $\pm$ 0.017 | 0.077 $\pm$ 0.039 | **0.042** $\pm$ 0.012 |
| | Correlated | 0.453 $\pm$ 0.231 | 0.147 $\pm$ 0.034 | 0.138 $\pm$0.039 | **0.004** $\pm$ 0.002 |
| | Spatially-varied | 0.023$\pm$0.008 | 0.241$\pm$0.043 | 0.218$\pm$0.034 | **0.012** $\pm$ 0.003 |

covariance $\mathbf{V}(\boldsymbol{\theta})$, which **alters the stationary distribution** (see Appendix A):

**Theorem 3.1** (informal). *For continuous-time MCLMC, the stationary distribution under anisotropic noise (if it exists) deviates from the target distribution.*

This is caused by a non-vanishing noise-induced drift term. In Table 1, we verify and quantify this effect by comparing the second-moment bias between samples and analytical expectations (Hoffman & Sountsov, 2022), $b^2 = (\bar{\boldsymbol{\theta}}^2_{\text{sample}} - \mathbb{E}[\boldsymbol{\theta}^2])^2/\text{Var}(\boldsymbol{\theta}^2)$, under different scenarios of explicit noise injection, based on the empirical average of squared sample $\bar{\boldsymbol{\theta}}^2_{\text{sample}}$. This leads to the important finding that the sample bias of SMILE-naive increases substantially across all settings under anisotropic noise.

**Noise Preconditioning** As shown above, stochastic MCLMC relies on isotropic gradient noise. However, in practice, the mini-batching covariance $\mathbf{V}(\boldsymbol{\theta})$ is often highly anisotropic, which can incur significant bias. To mitigate this, we apply local preconditioning to mini-batch gradients.

Suppose the local noise covariance $\mathbf{V}(\boldsymbol{\theta})$ is known and Cholesky decompose $\mathbf{V}(\boldsymbol{\theta}) = \mathbf{L}(\boldsymbol{\theta})\mathbf{L}(\boldsymbol{\theta})^\top$. We can define a local linear transformation of the space such that at the current position $\boldsymbol{\theta}_0$, the preconditioned coordinates are given by $\boldsymbol{\theta}' = \mathbf{L}(\boldsymbol{\theta}_0)^\top\boldsymbol{\theta}$. The preconditioned stochastic gradient in this local coordinate system is

$$\nabla_{\boldsymbol{\theta}'} \log p(\boldsymbol{\theta}'|\mathcal{B})|_{\boldsymbol{\theta}'=\mathbf{L}(\boldsymbol{\theta}_0)^\top\boldsymbol{\theta}_0} =$$
$$= \mathbf{L}(\boldsymbol{\theta}_0)^{-1}\nabla_{\boldsymbol{\theta}} \log p(\boldsymbol{\theta}|\mathcal{B})|_{\boldsymbol{\theta}=\boldsymbol{\theta}_0}$$
$$= \mathbf{L}(\boldsymbol{\theta}_0)^{-1}\left(B/N\nabla_{\boldsymbol{\theta}} \log p(\boldsymbol{\theta}_0|\mathcal{D}) + \sum_{i\in\mathcal{B}}\boldsymbol{\varepsilon}_{\mathcal{D}_i}\right).$$

By construction, the transformed noise term $\mathbf{L}(\boldsymbol{\theta}_0)^{-1}\sum_i\boldsymbol{\varepsilon}_{\mathcal{D}_i}$ has an identity covariance $\mathbf{I}$. This ensures that the noise scales are locally isotropic, reducing mini-batch gradient position-dependent anisotropy and thereby mitigating noise-induced drift.

Note that a spatially varying $\mathbf{L}(\boldsymbol{\theta})$ strictly requires a Riemannian correction term (involving the divergence of the preconditioner) to ensure the dynamics leave the posterior invariant (Girolami & Calderhead, 2011). However, following standard practice in large-scale SGMCMC (Li et al., 2016), we treat $\mathbf{L}(\boldsymbol{\theta})$ as locally constant. This approximation sidesteps the prohibitive computational cost of calculating the derivatives of the covariance matrix while significantly reducing the bias compared to unpreconditioned mini-batching.

In practice, estimating $\mathbf{V}(\boldsymbol{\theta})$ is computationally infeasible for large-scale models. Here, we propose to only use diagonal preconditioning based on moving averages to make computations tractable. Thus, the reparameterized variable is $\boldsymbol{\theta}' = \left(\sqrt{d}/\|\boldsymbol{\sigma}(\sum_{i\in\mathcal{B}}\boldsymbol{\varepsilon}_{\mathcal{D}_i})\|\right)\boldsymbol{\theta}\odot\boldsymbol{\sigma}(\sum_{i\in\mathcal{B}}\boldsymbol{\varepsilon}_{\mathcal{D}_i})$, where $\odot$ denotes the Hadamard product and we only need to estimate the gradient standard deviation $\boldsymbol{\sigma}(\sum_{i\in\mathcal{B}}\boldsymbol{\varepsilon}_{\mathcal{D}_i})$. The normalizing constant $\sqrt{d}/\|\boldsymbol{\sigma}(\sum_{i\in\mathcal{B}}\boldsymbol{\varepsilon}_{\mathcal{D}_i})\|$ is chosen such that $\boldsymbol{\theta}' = \boldsymbol{\theta}$ for isotropic noise. In addition to the estimation of the gradient standard deviation, we also require an estimate $\bar{\boldsymbol{g}}$ for the expected gradient, which is also computed using a moving average:

$$\bar{\boldsymbol{g}}^{(t+1)} \leftarrow (1-\alpha)\bar{\boldsymbol{g}}^{(t)} + \alpha\nabla_{\boldsymbol{\theta}} \log p(\boldsymbol{\theta}|\mathcal{B}),$$
$$\boldsymbol{\sigma}^{(t+1)} \leftarrow \sqrt{(1-\alpha)(\boldsymbol{\sigma}^{(t)})^2 + \alpha(\nabla_{\boldsymbol{\theta}} \log p(\boldsymbol{\theta}|\mathcal{B}) - \bar{\boldsymbol{g}}^{(t+1)})^2}.$$

$\alpha$ is a smoothing factor (fixed to $\alpha = 0.01$ throughout the paper) to ensure stable and low-variance estimates. We call this method preconditioned SMILE-naive (pSMILE-naive), as it preconditions the stochastic gradient to ensure the effective noise is isotropic. In principle, extra strong noise injection (Chen et al., 2014) and Fisher scoring (Ahn et al., 2012) would also be applicable to reduce the bias with potential sacrifice in efficiency.

**Analytical Benchmarks** Table 1 compares the squared bias $b^2$ of single-chain SMILE-naive, pSMILE-naive,

*Table 2.* Mean RMSE (↓) and LPPD (↑) results (± standard deviation across 3 train-test splits) for a 3 hidden-layer fully-connected neural network on regression tasks. All methods use 10 chains with different MAP solutions as starting points. Batch-wise sampling refers to the equivalent budget of sampling steps with respect to MILE (the gold standard), and epoch-wise sampling equalizes the number of passes through the dataset compared to MILE. Further experimental details can be found in Section E.1.

| Dataset | LPPD (↑) | | | RMSE (↓) | | |
|---|---|---|---|---|---|---|
| | Airfoil | Bikesharing | Energy | Airfoil | Bikesharing | Energy |
| **Full-batch** Gold Standard | | | | | | |
| MILE | $0.665 \pm 0.062$ | $0.226 \pm 0.043$ | $2.204 \pm 0.024$ | $0.152 \pm 0.014$ | $0.229 \pm 0.016$ | $0.032 \pm 0.002$ |
| **Stochastic** (Batch-wise Sampling) | | | | | | |
| SGHMC | $-0.176 \pm 0.023$ | $-0.092 \pm 0.029$ | $0.062 \pm 0.034$ | $0.265 \pm 0.020$ | $\mathbf{0.250 \pm 0.015}$ | $0.113 \pm 0.009$ |
| SMILE-naive | $0.280 \pm 0.008$ | $\mathbf{0.132 \pm 0.046}$ | $1.191 \pm 0.015$ | $\mathbf{0.185 \pm 0.006}$ | $0.236 \pm 0.017$ | $0.045 \pm 0.001$ |
| pSMILE-naive | $\mathbf{0.438 \pm 0.042}$ | $\mathbf{0.185 \pm 0.037}$ | $\mathbf{1.666 \pm 0.017}$ | $\mathbf{0.172 \pm 0.007}$ | $\mathbf{0.231 \pm 0.014}$ | $\mathbf{0.041 \pm 0.001}$ |
| **Stochastic** (Epoch-wise Sampling) | | | | | | |
| SGHMC | $0.177 \pm 0.037$ | $0.145 \pm 0.059$ | $1.063 \pm 0.020$ | $0.214 \pm 0.004$ | $\mathbf{0.236 \pm 0.019}$ | $0.053 \pm 0.002$ |
| SMILE-naive | $0.497 \pm 0.049$ | $0.167 \pm 0.058$ | $1.287 \pm 0.020$ | $0.166 \pm 0.007$ | $\mathbf{0.233 \pm 0.018}$ | $0.046 \pm 0.001$ |
| pSMILE-naive | $\mathbf{0.633 \pm 0.052}$ | $\mathbf{0.221 \pm 0.028}$ | $\mathbf{2.018 \pm 0.060}$ | $\mathbf{0.151 \pm 0.005}$ | $\mathbf{0.230 \pm 0.013}$ | $\mathbf{0.038 \pm 0.003}$ |

SGLD, and vanilla SGHMC across several anisotropic noise scenarios. In all settings, we assume the average noise magnitude is larger than that of the true gradient. We define three anisotropic noise scenarios based on the noise covariance matrix: (i) *Diagonal*, where the covariance matrix is a constant ill-conditioned diagonal matrix; (ii) *Correlated*, where the covariance matrix is the randomly rotated 'Diagonal' covariance; and (iii) *Spatially-varied*, where the covariance matrix is the 'Correlated' covariance times $\exp(-\theta_2/\sigma(\theta_2))$ with $\theta_2$ being the second element of $\boldsymbol{\theta}$.

The results show that although pSMILE-naive does not fully match the performance of SMILE-naive under ideal isotropic noise, it significantly reduces the bias in all three anisotropic settings, particularly for an ill-conditioned Gaussian (ICG) posterior. Furthermore, pSMILE-naive consistently outperforms both SGLD and SGHMC in these challenging scenarios. Beyond Gaussian mini-batch noise, we examined the performance and sensitivity of pSMILE-naive in various non-Gaussian noise settings in Section D.2, confirming that the performance under moderate violations of the Gaussian assumption is preserved but degraded under strong non-Gaussian noise. In addition, we evaluate pSMILE-naive on complex Gaussian mixtures in Section D.3, where it effectively traverses energy barriers to discover multiple modes.

### 3.3. A Naive SMILE in Practice

For a more challenging test, we evaluate our methods on a BNN regression benchmark where MILE is considered the current gold standard.

**The Benefit of Gradient Noise Preconditioning** We first run stochastic samplers using a *batch-wise sampling* approach, i.e., we produce one sample for every mini-batch step. This provides all samplers with the same number of total gradient computations as the MILE baseline. The results

(Table 2, first four rows) provide strong empirical support for our theory as both SMILE-naive and pSMILE-naive consistently outperform the SGHMC baseline in both log pointwise predictive density (LPPD) as a measure for the evaluation of the approaches' uncertainty quantification and root mean squared error (RMSE) to measure their prediction performance. Critically, pSMILE-naive markedly improves upon SMILE-naive, confirming that correcting for gradient noise anisotropy is crucial for the optimal performance of stochastic microcanonical dynamics.

**Closing the Gap to Full-batch MCMC** The batch-wise sampling comparison still implies that MILE makes more passes over the dataset than the stochastic samplers if both are run for the same number of iterations. To provide a fairer comparison, we also run *epoch-wise* sampling (Table 2, last three rows). This ensures that the samplers make the same number of full passes over the dataset as MILE. Under this condition, pSMILE-naive closes the remaining performance gap entirely, matching the performance of the full-batch MILE. This result is significant: it demonstrates that with proper preconditioning, stochastic microcanonical dynamics can achieve state-of-the-art sampling performance, effectively mitigating the performance gap often observed between stochastic and full-batch MCMC.

One downside is that the performance of these naive methods is rather sensitive to their hyperparameters. As shown in the Appendix (Figure 6), the step size crucially depends on the gradient noise, which in turn is related to the batch size. This sensitivity becomes more pronounced on more complex architectures. Further experiments on a LeNet architecture (62k parameters) for Fashion-MNIST classification (Table 6, Section E.2) reinforce the findings of the tabular benchmarks in Table 2. While SMILE-naive fails to sample meaningfully, pSMILE-naive performs exceptionally well, indicating that our gradient noise preconditioning remedy enables scaling to larger architectures.

**Pitfall:** Anisotropic mini-batch gradient noise introduces bias in MCLMC.
**Remedy:** Gradient noise preconditioning resolves the problem, facilitating state-of-the-art performance both in simulated examples and on small to medium-scale BNNs.

## 4. Scaling Stochastic Microcanonical Langevin

While gradient noise preconditioning resolves theoretical inconsistencies of stochastic microcanonical dynamics, a critical practical barrier remains: Naive implementations with fixed step sizes, performing very well for smaller models, break down when applied to modern architectures such as transformers. In the following, we discuss the cause of this problem and present our proposed solution, an adaptive tuning scheme that yields the algorithms (p)SMILE.

### 4.1. The Challenge of Scaling: Numerical Instability

When running previously proposed samplers in very high-dimensional and complex parameter spaces, performance notably degrades. Analyzing this result, we find that the microcanonical dynamic is very sensitive to the step size: a too large step size causes the sampler's trajectory to diverge rapidly, while one that is too small leads to prohibitively slow exploration.

This failure is demonstrated in Figure 1, where we apply SMILE-naive and pSMILE-naive to a ResNet-7 (428k parameters) on CIFAR-10. Despite preconditioning, tuning over various step sizes, and using cosine decay-type step size schedulers, both methods fail to produce meaningful results, diverging into unstable regions. In contrast, SGHMC remains stable, and our later introduced fully-tuned SMILE and pSMILE variants perform competitively. This highlights that simply correcting for gradient noise anisotropy is insufficient, and a robust tuning mechanism is needed.

**Pitfall:** The naive application of stochastic microcanonical Langevin Dynamics to large modern network architectures fails due to stability issues.
**Remedy:** Energy-variance-based numerical guardrails and adaptive step size scheduling enable robust and competitive scaling to large network architectures.

### 4.2. Energy Error-based Tuning

To address the numerical instabilities encountered at scale, a key challenge lies in managing errors introduced by the numerical integrator. As discussed in the background, the MCLMC dynamic provides a natural mechanism for this: while the ideal dynamic conserves total energy, any change in energy, $\Delta E$, serves as a direct proxy for the numerical error per integration step. This insight motivates a tuning

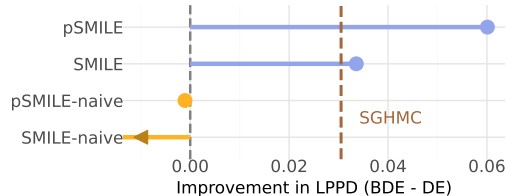

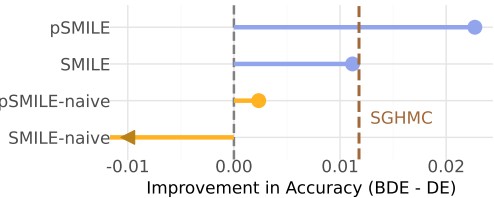

*Figure 1.* Differences between the Bayesian deep ensemble (BDE) performance of naive (orange) and tuned (blue) SMILE variants and a deep ensemble (DE) baseline for a ResNet-7 (428k parameters) on the CIFAR10 dataset. The x-axis is truncated at -0.01 for readability. For all samplers, we report the best performance of an ensemble of 8 chains over a grid of explored step sizes. Standard deviations over replications are comparable to those reported for the larger-scale setting reported in Table 9. A more detailed plot covering different step and batch sizes is given in Figure 8.

scheme based on monitoring the energy error $\Delta E$. Practically, it requires only a single user-specified hyperparameter (an initial step size, typically close to the optimizer's learning rate used in the warmstart), introduces no additional noise injection, and operates efficiently in high-dimensional SGMCMC regimes. The full procedure is given in Algorithm 1 and described below.

**Modeling Energy Error** To create robust guardrails and dynamic step-size adjustments, we need to assess whether a given energy error is typical or an outlier. This requires modeling the underlying distribution of the energy error to compute meaningful adaptive thresholds.

To this end, we model the distribution of $|\Delta E|$ in an online fashion using a Gamma distribution, which is well-suited for positive, skewed data. For this, exponential moving averages (EMAs) of its mean $\mu_{|\Delta E|}^{(t)}$ and standard deviation $\sigma_{|\Delta E|}^{(t)}$ are computed:

$$\mu_{|\Delta E|}^{(t+1)} \leftarrow (1-\beta)\mu_{|\Delta E|}^{(t)} + \beta|\Delta E|,$$
$$\sigma_{|\Delta E|}^{(t+1)} \leftarrow \sqrt{(1-\beta)(\sigma_{|\Delta E|}^{(t)})^2 + \beta(|\Delta E| - \mu_{|\Delta E|}^{(t+1)})^2},$$

where $\beta$ is the EMAs' smoothing parameter (set to 0.01 throughout the paper). To counteract the bias towards zero in the early stages of the tuning, we further apply a standard correction factor to the variance estimate, helping it converge more rapidly. In order to provide a tractable estimate for the energy error distribution, we then dynamically fit a Gamma distribution $\mathcal{G}a(\gamma_{\texttt{shape}}^{(t)}, \gamma_{\texttt{scale}}^{(t)})$ using moment-

**Algorithm 1** Energy variance-based tuning

**Require:** $\boldsymbol{\theta}^{(t)}, \boldsymbol{u}^{(t)}, \mu_{|\Delta E|}^{(t)}, \sigma_{|\Delta E|}^{(t)}, \beta, \kappa, a$ and $\delta$.

1: **Integrate** (as detailed in Section F.3):

$\boldsymbol{\theta}^{(t+1)}, \boldsymbol{u}^{(t+1)}, \Delta E \leftarrow \text{INTEGRATORSTEP}(\boldsymbol{\theta}^{(t)}, \boldsymbol{u}^{(t)}, \eta^{(t)})$.

2: Estimate the current Gamma distribution via moment matching (Equation (4)).

3: Update the exponential moving averages $\mu_{|\Delta E|}^{(t+1)}$ and $\sigma_{|\Delta E|}^{(t+1)}$ as detailed in Equation (3).

4: **Apply adaptive numerical guardrails** (Equation (5)).

5: **Adapt step size** $\eta^{(t)}$ according to Equation (6).

6: **Return** $\boldsymbol{\theta}^{(t+1)}, \boldsymbol{u}^{(t+1)}, \eta^{(t+1)}$

matching based on the empirical parameters

$$
\gamma_{\text{shape}}^{(t)} \leftarrow (\sigma_{|\Delta E|}^{(t)})^2 / \mu_{|\Delta E|}^{(t)}, \quad \gamma_{\text{scale}}^{(t)} \leftarrow \left( \mu_{|\Delta E|}^{(t)} / \sigma_{|\Delta E|}^{(t)} \right)^2.
\tag{4}
$$

Quantiles from this distribution are fitted in a highly practical online fashion, providing a reliable measure of extremity (see Section F.4 for details). Because the sampler begins from an optimized starting point (a high-likelihood region) with a meaningful step size (e.g., the final learning rate from the warmstart optimization), these errors are naturally contained within a reasonable range—meaning they are not excessive and do not significantly degrade performance. This creates a "golden window" for the EMA estimates to stabilize quickly (see Figure 9) and allows us to inform adaptive tuning as well as to define numerical guardrails throughout the sampling process.

**Numerical Guardrails** Numerical instability can cause the chain's trajectory to diverge, leading it into regions of extremely low likelihood from which recovery is computationally expensive and slow. To avoid this, we define a quantile threshold $\kappa$ based on a high quantile (0.98 by default) of the Gamma distribution. We approximate these quantiles efficiently using the Wilson-Hilferty transform (Wilson & Hilferty, 1931), denoting the quantile function by $\mathcal{G}a^{-1}(\cdot, \gamma_{\text{shape}}^{(t)}, \gamma_{\text{scale}}^{(t)})$. If $|\Delta E|$ exceeds this threshold, we reject the proposed step, resetting the position to its previous state and zeroing the momentum as

$$
(\boldsymbol{\theta}^{(t+1)}, \boldsymbol{u}^{(t+1)}) \leftarrow (\boldsymbol{\theta}^{(t)}, \mathbf{0}),
$$
$$
\text{if } |\Delta E| > \mathcal{G}a^{-1}(\kappa, \gamma_{\text{shape}}^{(t)}, \gamma_{\text{scale}}^{(t)}).
\tag{5}
$$

Our guardrail acts as a cost-efficient replacement for a Metropolis-Hastings correction step. While it differs in being non-probabilistic, we opted for this lightweight variant to leverage the distinct role of energy variance in MCLMC; in our large-scale BNN applications, a conventional adjustment would be computationally prohibitive and prone to extreme rejection rates (Garriga-Alonso & Fortuin, 2021). By preventing the chain from wasting computation in low-probability regions, we ensure a higher proportion of samples drawn from high-posterior density regions. Without

this guardrail, the sampler's performance catastrophically degrades (cf. Table 11).

**Adaptive Step Sizes** While the guardrails prevent catastrophic failure, a more nuanced mechanism can further boost efficient exploration. For this, we adapt the step size multiplicatively based on where $|\Delta E|$ lies relative to the Gamma quantiles: when errors are too large, the step sizes are decreased, and increased when they are unusually small (with defaults $\delta = 0.02, a = 0.1$):

$$
\eta^{(t+1)} = \begin{cases} \eta^{(t)}(1+\delta) & |\Delta E| < \tau_{\text{lo}}^{(t)} \\ \eta^{(t)}(1-\delta) & |\Delta E| > \tau_{\text{hi}}^{(t)} \\ \eta^{(t)} & \text{otherwise} \end{cases}
\tag{6}
$$

where $a$ parameterizes the probability of adaptation, $\tau_{\text{lo}}^{(t)} = \mathcal{G}a^{-1}(\frac{a}{3}, \gamma_{\text{shape}}^{(t)}, \gamma_{\text{scale}}^{(t)})$ and $\tau_{\text{hi}}^{(t)} = \mathcal{G}a^{-1}(1 - \frac{2a}{3}, \gamma_{\text{shape}}^{(t)}, \gamma_{\text{scale}}^{(t)})$. The update is asymmetric, giving stronger incentives to shrink the step size and thus biasing the dynamics toward stability. This design connects to adaptive MCMC (Andrieu & Éric Moulines, 2006; Haario et al., 2006; Roberts & Rosenthal, 2009), but is specialized for large-scale SGMCMC, where classical Metropolis-Hastings adjustments and detailed balance cannot be leveraged effectively (Garriga-Alonso & Fortuin, 2021).

### 4.3. Sampling of Contemporary Bayesian Neural Network Architectures

Equipped with our adaptive tuning scheme, we now evaluate (p)SMILE on contemporary BNN architectures, demonstrating their scalability and competitive performance.

**Image Classification** On CIFAR-10 using a ResNet-18 (11.2M parameters), pSMILE matches or outperforms both SGHMC and cSGLD across all metrics (Table 3). Single-chain results in Table 10 (Appendix) confirm that although (p)SMILE is configured for a single exploration-exploitation cycle per chain, it remains highly effective in isolation. These results match cSGLD's performance without requiring its characteristic aggressive within-chain cycling. Ultimately, pSMILE yields the best overall balance of accuracy and uncertainty quantification, demonstrating the synergistic benefit of our proposed remedies.

Robustness is further validated on a 22M-parameter Vision Transformer using Imagenette, where pSMILE maintains its competitive advantage (cf. Table 3). This consistent performance across distinct, high-dimensional architectures generalizes our findings for large-scale inference.

**Language Modeling** In the experiments corresponding to Figure 2 and Table 14 (Section E.6), we test robustness on a nanoGPT model (10.8M parameters, Karpathy, 2022).

*Table 3.* Large-scale image classification results for two tasks comparing different ensemble-based methods (rows) and performance metrics (columns). For details, single chain performance and standard errors see Section E and in particular Tables 9, 10 and 13.

| Task / Model | Method | LPPD (↑) | Acc. (↑) | F1 (↑) | Brier (↓) | NLL (↓) | AUROC (↑) | AURC (↓) |
|---|---|---|---|---|---|---|---|---|
| CIFAR-10 / ResNet-18 (11.2M) | DE | −0.3026 | 0.8995 | 0.8938 | 0.1545 | 0.4507 | 0.9934 | 0.0161 |
| | cSGLD | **−0.2584** | **0.9140** | **0.9139** | **0.1264** | 0.4059 | **0.9954** | **0.0111** |
| | SGHMC | −0.2908 | 0.9104 | 0.9103 | 0.1389 | 0.4942 | 0.9949 | 0.0127 |
| | SMILE | −0.2763 | 0.9101 | 0.9100 | 0.1344 | 0.4071 | 0.9951 | 0.0122 |
| | **pSMILE** | −0.2598 | 0.9135 | 0.9134 | **0.1264** | **0.3900** | **0.9954** | 0.0113 |
| Imagenette / ViT (22M) | DE | −0.9858 | 0.7688 | 0.7702 | 0.3367 | 1.4957 | 0.9691 | 0.0645 |
| | cSGLD | −0.7815 | 0.7638 | 0.7640 | 0.3325 | 3.4947 | 0.9645 | 0.0691 |
| | SGHMC | −0.7841 | 0.7578 | 0.7582 | 0.3440 | 1.2928 | 0.9625 | 0.0758 |
| | **pSMILE** | **−0.7708** | **0.7732** | **0.7738** | **0.3112** | **1.1985** | **0.9712** | **0.0589** |

We ablate the initial step size over four orders of magnitude. For reference, the well-working learning rate of AdamW for the DE optimization is $2 \cdot 10^{-4}$, which also coincides with the best performing step size for all considered SGMCMC samplers. The results reveal robustness as a key strength of our method. While SGHMC's performance degrades catastrophically with a misspecified step size (cf. Figure 2), pSMILE consistently improves upon the strong DE baseline across all tested step sizes. This increased robustness to the initial step size is a significant practical advantage, reducing the need for costly hyperparameter sweeps common in SGMCMC (see Section E for further practical tuning advice).

### 4.4. Analysis and Ablations of the Tuning Mechanism

We conduct a series of ablations to better understand the behavior of our proposed adaptive scheme. Our hyperparameter robustness analyses (Tables 11 and 12 in the Appendix) show that the energy-variance-based tuning is robust across a range of settings. Notably, the ablation on our guardrail mechanism (Table 11) highlights its necessity: without it, naive sampling fails catastrophically. The influence of the

batch size (Figure 7, Appendix) is also explored; while larger batches improve performance, the gains diminish, confirming that our method remains effective with moderate batch sizes. Finally, our analyses of the tuning dynamics (Figures 9 and 10) reveal that the adaptive step size and Gamma distribution parameters rapidly converge to a stable regime. This is a key user-friendly feature, as it heavily reduces the need for manual tuning, which poses a non-trivial challenge in many traditional MCMC implementations (e.g., consider the hyperparameters of the cyclical schedule to be specified in cSGLD). Furthermore, the empirically observed energy errors appear to be well described by the proposed Gamma distribution (Section E.7).

> **Takeaway:** 1) Gradient noise preconditioning in combination with 2) energy variance-based tuning enables the robust and successful application of stochastic microcanonical Langevin dynamics to modern BNN architectures, consistently yielding state-of-the-art performance.

## 5. Discussion

This work bridges the gap between the strong performance of full-batch microcanonical Langevin Monte Carlo and the scalability needs of Bayesian inference. We show that naive stochastic adaptations fail due to gradient noise bias and instability, and propose two key solutions: a principled preconditioning scheme for correctness, and an energy-variance-based tuner for stability. Our resulting method, (p)SMILE, matches and often outperforms strong baselines, demonstrating that the benefits of microcanonical dynamics can be realized in the stochastic regime. This establishes (p)SMILE as a practical and powerful SGMCMC method for complex models like BNNs. All experiments and our software are made publicly available at https://github.com/EmanuelSommer/SMILE.

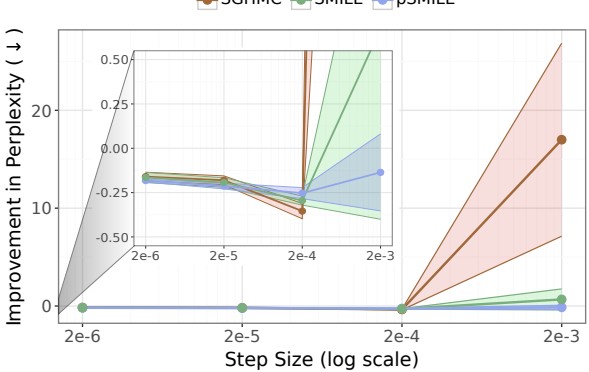

*Figure 2.* Robustness assessment: Perplexity improvement (smaller is better, std. dev. as shaded area) of MCMC sampling over the optimized warmstart across samplers and step sizes for the nanoGPT model with 10.8M parameters on modern-shakespeare.

**Limitations and Future Work** Our method relies solely on mini-batch gradient noise to drive dynamics. While effective, we did not study reintroducing explicit noise as

in the original MCLMC SDE, which could enhance exploration but adds further tuning complexity. Exploring this trade-off is a promising future direction. SMILE relies on a linear, diagonal preconditioner estimated via a dynamic procedure. While this effectively scales the dynamics, it entails a residual approximation bias. This is an inherent, shared constraint of stochastic gradient-based samplers where exact correction is generally intractable (Wenzel et al., 2020).

## Acknowledgements

This research was supported by grants from NVIDIA and utilized NVIDIA products, including NVIDIA A100 and NVIDIA RTX 6000 GPUs. DR's research is funded by the Deutsche Forschungsgemeinschaft (DFG, German Research Foundation) – 578966082. This material is also based upon work supported in part by the Heising-Simons Foundation grant 2021-3282, by the NSF CSSI grant award No. 2311559, and by the U.S. Department of Energy, Office of Science, Office of Advanced Scientific Computing Research under Contract No. DE-AC02-05CH11231 at Lawrence Berkeley National Laboratory to enable research for Data-intensive Machine Learning and Analysis. The authors thank Julius Kobialka for helpful discussions and also thank the four anonymous ICML 2026 reviewers for their constructive and helpful feedback.

## Impact Statement

This paper presents work whose goal is to advance the field of Machine Learning. There are many potential societal consequences of our work, none of which we feel must be specifically highlighted here.

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

# A. Stationary Distribution in the Continuous-time Limit

## A.1. Full-batch

We start by reviewing the full-batch continuous-time MCLMC. We denote the variables as $\boldsymbol{z} = (\boldsymbol{\theta}, \boldsymbol{u})$, where $\boldsymbol{z} \in \mathbb{R}^d \times S^{d-1}$, i.e., the velocity is normalized to the unit sphere. We will use Greek letter indices to denote parameters on the sphere and Latin letter indices to denote parameters in the Euclidean space. We will adopt Einstein convention, which implies a sum whenever there are repeated upper and lower indices. We raise and lower Greek indices using the metric $g_{\mu\nu}$ and its inverse $g^{\mu\nu}$, e.g. $\partial^\mu := g^{\mu\nu}\partial_\nu$.

MCLMC dynamics can be expressed as a Stratonovich degenerate diffusion on the manifold,

$$\mathrm{d}\boldsymbol{z} = B(\boldsymbol{z})\,\mathrm{d}t + \eta \sum_{i=1}^d \sigma_i(\boldsymbol{z}) \circ \mathrm{d}W_i. \tag{7}$$

The notation convention is as in Robnik & Seljak (2024), and we briefly summarize the notation here: $W_i$ are independent $\mathbb{R}$-valued Wiener processes, $\circ$ denotes that the SDE is to be interpreted in the Stratonovich sense, $\sigma_i$ are vector fields, in coordinates expressed as

$$\sigma_i(\boldsymbol{\vartheta}) = g^{\mu\nu}(\boldsymbol{\vartheta}) \frac{\partial u_i}{\partial \vartheta^\nu}(\boldsymbol{\vartheta}) \frac{\partial}{\partial \vartheta^\mu}(\boldsymbol{\vartheta}). \tag{8}$$

Given that $\boldsymbol{u}$ lives in the unit sphere, $\boldsymbol{u}$ are parametrized with spherical coordinates,

$$\boldsymbol{u}(\boldsymbol{\vartheta}) = (\cos\vartheta_1,\, \sin\vartheta_1\cos\vartheta_2,\, ...,\, \sin\vartheta_1\sin\vartheta_2...\cos\vartheta_{d-1},\, \sin\vartheta_1\sin\vartheta_2...\sin\vartheta_{d-1}), \tag{9}$$

and we denote $\boldsymbol{\vartheta} = (\vartheta_1, \vartheta_2, ..., \vartheta_{d-1})$. The metric tensor on the sphere is

$$g_{\mu\nu} = \sum_{i=1}^d \partial_\mu u_i(\vartheta)\partial_\nu u_i(\vartheta). \tag{10}$$

The drift vector field is

$$B(\boldsymbol{\theta}, \boldsymbol{u}) = (\boldsymbol{u},\, \mathbf{P}(\boldsymbol{u})\nabla_{\boldsymbol{\theta}} \log p(\boldsymbol{\theta}|\mathcal{D})/(d-1)), \tag{11}$$

and $\mathbf{P}(\boldsymbol{u}) = \mathbf{I} - \boldsymbol{u}\boldsymbol{u}^\top$ is the projection tensor.

The Fokker-Planck equation corresponding to Equation 7 is

$$\dot{\rho} = -\left(\nabla_i(\rho B^i) + \nabla_\mu(\rho B^\mu)\right) + \frac{1}{2}\nabla_\mu\nabla_\nu(D^{\mu\nu}\rho), \tag{12}$$

where $\nabla$ is the covariant derivative, and the diffusion tensor is

$$D^{\mu\nu} = \eta^2 \sum_{i=1}^d \sigma_i^\mu \sigma_i^\nu = \eta^2 \sum_{i=1}^d \partial^\mu u_i\, \partial^\nu u_i = \eta^2 g^{\mu\nu}, \tag{13}$$

so the diffusion term in the Fokker-Planck equation becomes the Laplacian. It is shown in Robnik & Seljak (2024) that the stationary distribution of the Fokker-Planck equation has the form of $\rho_\infty \propto p(\boldsymbol{\theta}|\mathcal{D})\sqrt{g}$ because both terms on RHS of Equation (12) vanish with $\rho_\infty$. Here $g$ is the determinant of the metric.

## A.2. Isotropic Mini-batching Noise

Now we consider mini-batching noise on the gradient. We are interested in the limit of the step size going to zero. Mini-batching noise can then be modeled as a Wiener process.

Note, however, that in the limit, the process converges to the Itô's SDE, not Stratonovich's SDE. This is because the realization of the mini-batching noise is always taken at the beginning of the step. For Gaussian mini-batching noise with isotropic covariance matrix, $\boldsymbol{\Sigma} = \eta^2 \mathbf{I}$, we thus obtain

$$\mathrm{d}\boldsymbol{z} = B(\boldsymbol{z})\,\mathrm{d}t + \eta \sum_{i=1}^d \sigma_i(\boldsymbol{z})\,\mathrm{d}W_i, \tag{14}$$

which is equivalent to the following Stratonovich SDE:

$$d\boldsymbol{z} = B_{\text{SG}}(\boldsymbol{z})\, dt + \eta \sum_{i=1}^{d} \sigma_i(\boldsymbol{z}) \circ dW_i, \tag{15}$$

where the drift has an additional term:

$$B_{\text{SG}} = B - \frac{1}{2} \sum_{i=1}^{d} \nabla_{\sigma_i} \sigma_i. \tag{16}$$

**Lemma 1.** $\sigma_i$ *are geodesic vector fields. Specifically,*

$$\nabla_{\sigma_i} \sigma_i = \lambda_i(\boldsymbol{\vartheta}) \sigma_i \qquad \lambda_i(\boldsymbol{\vartheta}) = -u_i(\boldsymbol{\vartheta}).$$

*Proof.* To calculate the parallel transport of the vector field along itself, we will use the Gauss formula, by which the Levi–Civita connection of a submanifold is the tangential projection of the ambient derivative. Viewing the sphere $S^{d-1}$ as a submanifold of the Euclidean space $\mathbb{R}^d$, we can use this to calculate

$$\nabla_X Y = \mathbf{P}\mathbf{D}[X]Y,$$

where $X$ and $Y$ are any smooth vector fields in the Euclidean space, which are tangential to the sphere when restricted to the sphere. $\mathbf{P} = (\mathbf{I} - \boldsymbol{u}\boldsymbol{u}^T)$ here again is the projection operator and $\mathbf{D}[X]$ denotes the Jacobian matrix. $\sigma_i = \mathbf{P}\boldsymbol{e}_i$, where $\boldsymbol{e}_i$ is a unit vector in direction $i$ is such a vector field. Its Jacobian is

$$D[\sigma_i] = -\mathbf{I}(\boldsymbol{e}_i \cdot \boldsymbol{u}) - \boldsymbol{u} \otimes \boldsymbol{e}_i,$$

where $\otimes$ is the Kronecker product. Using the projected ambient derivative gives:

$$\nabla_{\sigma_i} \sigma_i = -\mathbf{P}(\mathbf{I}(\boldsymbol{e}_i \cdot \boldsymbol{u}) + \boldsymbol{u} \otimes \boldsymbol{e}_i)\mathbf{P}\boldsymbol{e}_i = -(\boldsymbol{e}_i \cdot \boldsymbol{u})\sigma_i,$$

where we have used that $\mathbf{P}^2 = \mathbf{P}$ and $\mathbf{P}(\boldsymbol{u} \otimes \boldsymbol{e}_i) = 0$. $\qquad\square$

Using the result from this Lemma, we verify that the noise-induced drift term vanishes in the isotropic case:

$$[B_{\text{SG}}]_\mu = B_\mu - \frac{1}{2} \sum_{i=1}^{d} [\nabla_{\sigma_i} \sigma_i]_\mu = B_\mu + \frac{1}{2} \sum_{i=1}^{d} u_i \frac{\partial u_i}{\partial \vartheta^\mu} = B_\mu + \frac{1}{2} \frac{\partial}{\partial \vartheta^\mu} \sum_{i=1}^{d} u_i^2 = B_\mu. \tag{17}$$

This leads to the following formalization of the properties of MCLMC under isotropic stochastic gradients:

**Proposition 1** (Stationarity under Isotropic Noise). *If the mini-batching noise can in the limit be modeled as an isotropic Wiener process, the additional drift term $\frac{1}{2} \sum \nabla_{\sigma_i} \sigma_i$ vanishes exactly. Consequently, the Fokker-Planck equation (Eq. 12) remains invariant, preserving the target stationary distribution $p(\boldsymbol{\theta}|\mathcal{D})$ and the geometric ergodicity of the original dynamics.*

### A.3. Anisotropic Mini-batching Noise

Let the mini-batching noise now have some general covariance matrix $\boldsymbol{\Sigma}$. Without loss of generality, we may assume that $\boldsymbol{\Sigma} = \text{Diag}(\eta_1^2, \eta_2^2, \ldots, \eta_d^2)$, by rotating the coordinate system if it is not in this form. We now get

$$d\boldsymbol{z} = B_{\text{SG}}(\boldsymbol{z})\, dt + \sum_{i=1}^{d} \eta_i \sigma_i(\boldsymbol{z}) \circ dW_i, \tag{18}$$

and

$$B_{\text{SG}} = B - \frac{1}{2} \sum_{i=1}^{d} \eta_i^2 \nabla_{\sigma_i} \sigma_i = B + \frac{1}{2} \sum_{i=1}^{d} \eta_i^2 u_i \sigma_i, \tag{19}$$

where we have used the Lemma from the previous section. The diffusion tensor is now

$$D_{\text{SG}}^{\mu\nu} = \sum_{i=1}^{d} \eta_i^2 \sigma_i^\mu \sigma_i^\nu = \sum_{i=1}^{d} \eta_i^2 \partial^\mu u_i \, \partial^\nu u_i. \tag{20}$$

Putting these together, the Fokker-Planck equation becomes

$$\dot{\rho} = - \left( \nabla_i(\rho B^i) + \nabla_\mu(\rho B^\mu) \right) + \frac{1}{2} \sum_{i=1}^{d} \eta_i^2 \left( -\nabla_\mu(\rho u_i \sigma_i^\mu) + \nabla_\mu \nabla_\nu(\rho \partial^\mu u_i \partial^\nu u_i) \right). \tag{21}$$

By analyzing this modified equation, we establish the formal statement for Theorem 3.1 in the main text:

**Theorem 3.1.** *For continuous-time MCLMC, the stationary distribution under anisotropic noise (if it exists) deviates from the target distribution. Specifically, because the noise-induced drift $\frac{1}{2} \sum \eta_i^2 u_i \sigma_i$ does not vanish for non-uniform $\eta_i$, the target posterior no longer satisfies the stationary Fokker-Planck equation.*

### A.4. Finite Step Size, Distribution-Free Sketch via a Discrete Generator

While our formal results characterize the continuous-time dynamics, developing a complete finite-step-size theory for the discretized dynamics remains an important yet highly non-trivial direction for future work (see, e.g., Durmus et al., 2019; Wang et al., 2025). Nevertheless, we provide a sketch of a discrete analysis in the following, using finite-step analysis analogous to Section 3.1 in Vollmer et al. (2016).

The continuous-time analysis above captures the limit in which mini-batch noise can be treated as a Wiener process. In practice, however, we can use a discrete integrator with finite step size $\varepsilon$, and this discretization can introduce a systematic $\mathcal{O}(\varepsilon)$ perturbation of the dynamics.

To obtain a distribution-free intuition for this finite-step bias, we can consider the corresponding discrete-time Markov chain through its discrete generator. A second-order Taylor expansion shows that, at leading order $\mathcal{O}(\varepsilon)$, the influence of the mini-batch noise enters through its *covariance structure* rather than through the full noise distribution, with higher-order moments (such as skewness and kurtosis) entering only at $\mathcal{O}(\varepsilon^2)$ and beyond. Moreover, because the velocity is constrained to the microcanonical sphere, the bias mechanism can be interpreted as a form of *diffusion anisotropy on the sphere*: when the noise covariance is not effectively isotropic in tangent directions, the discrete dynamics develops a noise-induced drift that perturbs the stationary distribution.

If the effective noise is (approximately) isotropic, the perturbation reduces to a spherical diffusion that eliminates the noise-induced drift, leaving only standard deterministic integrator errors and higher-order stochastic terms; if it is anisotropic, stationarity is generally distorted at leading order in $\varepsilon$. These considerations motivate local preconditioning as a practical way to suppress the anisotropic part of the noise covariance, thereby reducing the dominant bias.

A complete, fully rigorous finite-step-size derivation for the specific integrators used in our method is technically involved and is therefore left for future work.

## B. Experimental Setup and General Details

### B.1. Software and Computing Environment

Experiments were implemented in Python using `jax` (Bradbury et al., 2018), `BlackJAX` (Cabezas et al., 2024), and extensions of the codebase of Sommer et al. (2025), with selected baselines from `posteriors` (Duffield et al., 2025). Computations were performed on two NVIDIA RTX A6000 or four NVIDIA A100 GPUs and a 64-core AMD Ryzen™ Threadripper™ CPU; CPU parallelism was used for smaller tasks, while large models like CNNs were trained on GPUs. A comprehensive codebase is available at `https://github.com/EmanuelSommer/SMILE`.

### B.2. Datasets & Optimization

Table 4 summarizes the benchmark datasets utilized in our BNN experiments. For all tabular benchmarks, unless specified otherwise, we use a 70% train, 10% validation, and 20% test split together with a fully connected model architecture

of three hidden layers with 16 neurons each. For image classification benchmarks, we adopt the standard train/test split and employ CNN/ResNet-type architectures of varying size. Before training the nanoGPT model, we translated the `tiny-shakespeare` dataset available from (Karpathy, 2022) into more modern English using Gemini 2.5 Pro to facilitate a more accessible assessment of the quality of the generated text. This was done using the prompt "`Please translate the attached file into simplified modern English. Keep the structure of the text as is (new line for each speaker, speaker name followed by a colon, then the sentence in a new line)`", together with an attached `txt` file of the original text. We call this dataset `modern-shakespeare` and provide it within our public code repository for reproducibility. Furthermore, we use ADAM with decoupled weight decay (Loshchilov & Hutter, 2019) for all DEs and vanilla optimizations. Unless stated otherwise, we assume a standard Gaussian prior, $\mathcal{N}(\mathbf{0}, \boldsymbol{I}_d)$, as is common practice (Kobialka et al., 2026).

*Table 4.* Datasets used in the Bayesian Deep Learning experiments.

| Dataset | Size | Features | Source |
|---|---|---|---|
| Airfoil | 1503 | 5 | (Dua & Graff, 2017) |
| Bikesharing | 17379 | 13 | (Fanaee-T, 2013) |
| Energy | 768 | 8 | (Tsanas & Xifara, 2012) |
| F(ashion)-MNIST | 60000 | 28x28 | (Xiao et al., 2017) |
| CIFAR-10 | 60000 | 28x28 | (Krizhevsky et al., 2009) |
| `modern-shakespeare` | 39890 (rows) | 65 (CharacterTokenizer) | adapted from (Karpathy, 2022) |
| Imagenette | 13394 | 160x160 | (Howard, 2019) |

### B.3. Evaluation

We evaluate predictive performance using a range of metrics, with specific metrics chosen based on the task type (e.g., classification, regression, or language modeling). For evaluating the quality of the full predictive distribution and uncertainty, we utilize the log pointwise predictive density (LPPD) on a held-out test set $\mathcal{D}_{\text{test}}$, as advocated by Gelman et al. (2014). The LPPD is defined as:

$$\text{LPPD} = \frac{1}{n_{\text{test}}} \sum_{(\boldsymbol{y}^*, \boldsymbol{x}^*) \in \mathcal{D}_{\text{test}}} \log \left( \frac{1}{K \cdot S} \sum_{k=1}^{K} \sum_{s=1}^{S} p\left(\boldsymbol{y}^* | \boldsymbol{\theta}^{(k,s)}(\boldsymbol{x}^*)\right) \right) \tag{22}$$

Where $\boldsymbol{\theta}^{(k,s)}$ are the obtained posterior samples from $K$ chains of $S$ samples each. This metric measures how well the predictive distribution covers the observed labels, with higher values indicating a better fit.

We note that, particularly in Bayesian neural networks, high LPPD does not necessarily imply faithful posterior approximation in function space due to the singular and non-identifiable nature of the posterior (Deshpande et al., 2024). We therefore interpret LPPD jointly with complementary predictive and calibration metrics.

For classification tasks, we assess point predictions using accuracy, which measures the proportion of correct predictions. Following Zhou et al. (2025) we also consider the Brier score, which quantifies the mean squared error of the predicted probabilities, and the Area Under the Receiver Operating Characteristic (AUROC), which evaluates the model's discriminative ability. Furthermore, we examine calibration with the Area Under the Reliability Curve (AURC).

For regression tasks, the Root Mean Squared Error (RMSE) is employed to evaluate the accuracy of point predictions.

Further, for models, such as our nanoGPT model, we report the Negative Log-Likelihood (NLL) or Perplexity to assess how well the model predicts the test data. Perplexity, a common metric for language models, is a function of the average NLL (more specifically Perplexity := $\exp(\text{NLL})$) and indicates the effective number of choices the model has at each step, with a lower value representing better performance. The specific metrics used in each experiment are indicated in the respective results sections.

### B.4. LLM Usage

The only use of Large Language Models (LLMs) was for minor language, grammar, and stylistic edits, as well as trivial coding support (such as plotting scripts). No part of the scientific work or core implementation was generated by LLMs.

## C. Analytical Benchmark

In Table 1 we use four SGMCMC samplers, SMILE-naive , SGLD, vanilla SGHMC, and pSMILE-naive, to sample three 10-dimensional analytical posteriors with explicit noise injection. The three analytical posteriors are

1. **Ill-Conditioned Gaussian (ICG)** The distribution is $\mathcal{N}(0, \boldsymbol{\Sigma} = \mathbf{R}^\top \boldsymbol{\Lambda} \mathbf{R})$, where $\boldsymbol{\Lambda}$ is a diagonal matrix with eigenvalues equally sampled in log space from $1/100$ to $100$ and $\mathbf{R}$ is a random rotation matrix.

2. **Rosenbrock** This is a product of 5 banana-shaped posterior from Grumitt et al. (2022). The posterior is $\log p(\boldsymbol{\theta}) = -\sum_{i=1}^{d/2}[(\theta_{2i-1}^2 - \theta_{2i})^2/Q + (\theta_{2i-1} - 1)^2]$ with $Q = 0.1$ in our setting.

3. **Neal's Funnel** This is a hierarchical model with $\theta_1 \sim \mathcal{N}(0, 3)$ and $\theta_i \sim \mathcal{N}(0, e^{\theta/2})$, $i \in [2, ..., 10]$.

The explicit noise injection is realized by adding a noise term in the $\log p(\boldsymbol{\theta})$,

$$\log p(\boldsymbol{\theta})_{\text{noise}} = \log p(\boldsymbol{\theta}) + \boldsymbol{\epsilon}^\top \boldsymbol{\theta} \tag{23}$$

where $\boldsymbol{\epsilon}$ is a 10-dimensional Gaussian noise with covariance matrix $\mathbf{V}(\boldsymbol{\theta})$. In 'Isotropic' case, $\mathbf{V}_{\text{iso}} = 256\mathbf{I}$; in 'Diagonal' case, $\mathbf{V}_{\text{diag}} = \mathbf{V}_{\text{iso}}\boldsymbol{\Lambda}$ and the $\boldsymbol{\Lambda}$ is the same one used in ICG posterior; in 'Correlated' case, $\mathbf{V}_{\text{corr}} = \mathbf{R}'^\top \mathbf{V}_{\text{diag}} \mathbf{R}'$ and $\mathbf{R}'$ is another random rotation matrix different from the rotation matrix used in ICG posterior; in 'Spatially-Varied' case, the covariance matrix is $\mathbf{V}_{\text{spa}}(\boldsymbol{\theta}) = \mathbf{V}_{\text{corr}} \exp(-\theta_2/\sigma(\theta_2))$, with $\theta_2$ being the second element of $\boldsymbol{\theta}$. In each scenario, we initialize the position at the mean of the posterior and run 10 chains for $10^6$ samples.

The optimal step size for all SGMCMC samplers is determined from a grid search. We first run benchmark for $\Delta t = 10^i$, $i \in \{-6, -5, ..., 0\}$, then for optimal $i_{\text{opt}}$ we evaluate the performance for 15 step sizes spaced equally from $i_{\text{opt}} - 1$ to $i_{\text{opt}} + 1$ and report the bias in Table 1. For ICG and Rosenbrock, the **average bias** over all 10 dimensions is reported, and we use **maximum bias** for Neal's Funnel, because in practice the bias of $\theta_0$ is significantly larger than other parameters. We employ a bootstrapping technique to estimate the standard deviation of the mean bias by repeatedly drawing 10 chains with replacement from the original 10 chains, and report the standard deviation in Table 1.

## D. Further Results on Analytical Targets

Here we provide a collection of additional analytical benchmarks, validating our design choices and pSMILE-naive performances.

### D.1. Energy Error Distribution

To assess whether the numerical guardrail of Section 4.2 is needed on the 10-dimensional analytical targets, we examined the per-step energy-error distribution observed when sampling the benchmarks of Table 1. As a representative example, the empirical $|\Delta E|$ distribution on the ICG target is shown in Figure 3. Its tail is lighter than both the moment-matched Gamma and the moment-matched Half-Normal fits, and excursions beyond the 98th-percentile of the Gamma fit (the default guardrail threshold) are extremely rare. We therefore omit the guardrail on the analytical benchmarks, and use pSMILE-naive (without energy-error tuning) for analytical benchmarks.

### D.2. Non-Gaussian Noise Experiments

The Gaussian-gradient-noise assumption implied by the central limit theorem can fail in deep learning, particularly under small batch sizes or heavy-tailed data (Simsekli et al., 2019). We therefore stress-test pSMILE-naive by replacing the Gaussian injected noise in Table 1 with two symmetric heavy-tailed distributions: the Laplacian and the Student's-$t$ ($\nu = 5$). All other settings are identical to those in Table 1; the resulting biases are reported in Table 5.

For each step, we first draw a vector $\boldsymbol{\epsilon} = (\epsilon_1, \ldots, \epsilon_{10})$ whose components are i.i.d. samples from the chosen distribution, scaled to unit variance ($\sigma(\epsilon_i) = 1$). For each covariance structure $\text{s} \in \{\text{iso}, \text{aniso}, \text{corr}, \text{spa}\}$, the injected noise is then

$$\boldsymbol{\epsilon}_{\text{s}} = \mathbf{L}_{\text{s}} \boldsymbol{\epsilon}, \tag{24}$$

where $\mathbf{L}_{\text{s}}$ is the Cholesky factor of the corresponding covariance matrix $\mathbf{V}_{\text{s}}$ defined in Section C.

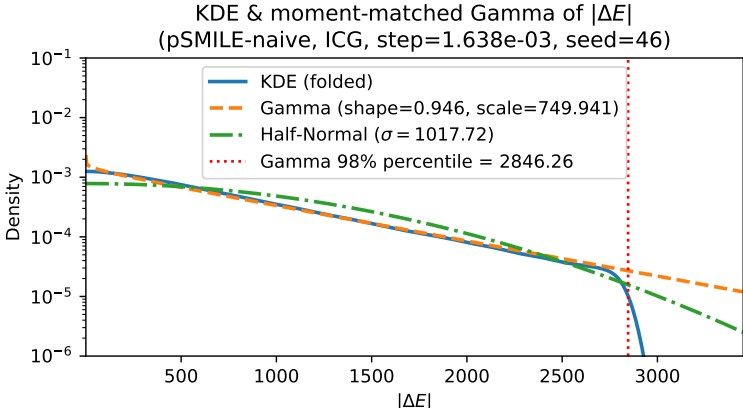

*Figure 3.* The KDE plot of the absolute energy error of pSMILE-naive on the ill-conditioned Gaussian target. **Blue**: KDE of $|\Delta E|$ across all sampling steps. **Orange**: moment-matched Gamma distribution used in Section 4.2. **Green**: Half-Normal distribution with matched standard deviation. **Red**: 98th-percentile of the fitted Gamma, the default threshold of the numerical guardrail in Section 4.2.

*Table 5.* Bias $b^2(\downarrow)$ of pSMILE-naive on three 10-$d$ analytical posteriors with explicitly injected Gaussian and Heavy-tail noise. The target is the same as Table 1.

| Target | Sub-type | Gaussian | Student's-t | Laplacian |
|---|---|---|---|---|
| **ICG** | Isotropic | $0.006 \pm 0.001$ | $0.004 \pm 0.001$ | $0.004 \pm 0.001$ |
| | Anisotropic | $0.038 \pm 0.008$ | $0.071 \pm 0.020$ | $0.072 \pm 0.020$ |
| | Correlated | $0.055 \pm 0.007$ | $0.078 \pm 0.010$ | $0.126 \pm 0.054$ |
| | Spatially-varied | $0.093 \pm 0.019$ | $0.083 \pm 0.015$ | $0.081 \pm 0.011$ |
| **Rosenbrock** | Isotropic | $0.004 \pm 0.001$ | $0.007 \pm 0.001$ | $0.009 \pm 0.001$ |
| | Anisotropic | $0.046 \pm 0.002$ | $0.045 \pm 0.002$ | $0.046 \pm 0.002$ |
| | Correlated | $0.070 \pm 0.005$ | $0.063 \pm 0.028$ | $0.064 \pm 0.009$ |
| | Spatially-varied | $0.052 \pm 0.005$ | $0.068 \pm 0.019$ | $0.049 \pm 0.009$ |
| **Funnel** | Isotropic | $0.021 \pm 0.005$ | $0.016 \pm 0.004$ | $0.022 \pm 0.002$ |
| | Anisotropic | $0.042 \pm 0.012$ | $0.039 \pm 0.009$ | $0.063 \pm 0.009$ |
| | Correlated | $0.004 \pm 0.002$ | $0.036 \pm 0.009$ | $0.019 \pm 0.006$ |
| | Spatially-varied | $0.012 \pm 0.003$ | $0.084 \pm 0.027$ | $0.061 \pm 0.019$ |

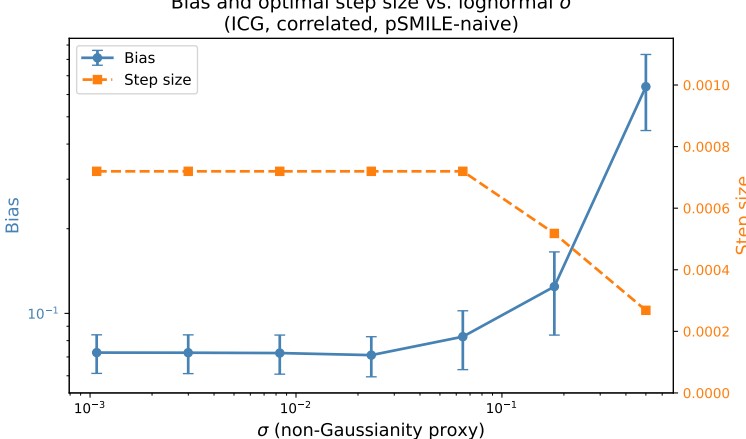

*Figure 4.* Minimal bias (blue) and the corresponding optimal step size (orange) of pSMILE-naive on the ill-conditioned Gaussian target, as a function of the log-normal noise parameter $\sigma$.

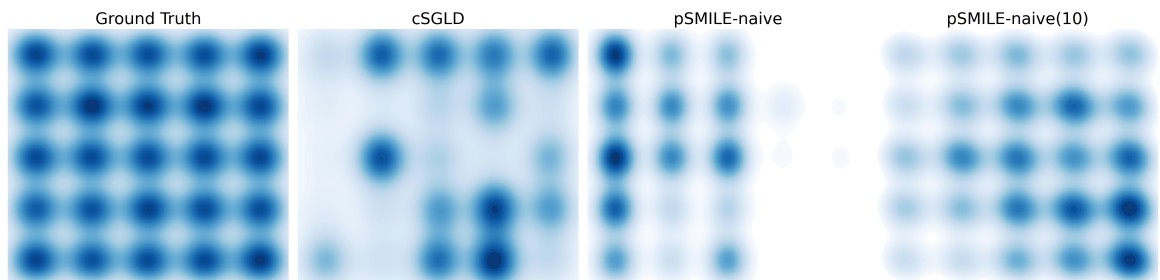

*Figure 5.* Posterior samples for the first two dimensions of a 10-dimensional Gaussian Mixture Model. **Left:** Ground truth density. **Middle Left:** Single-chain cSGLD samples. **Middle Right:** Single-chain pSMILE-naive samples. **Right:** pSMILE-naive samples using 10 independent chains. Both methods were tuned to have a matching average update norm per step. pSMILE-naive successfully traverses modes even in the single-chain setting.

Across all three targets and four covariance structures, the bias under heavy-tailed noise is comparable to and at worst mildly elevated over the Gaussian case. Crucially, pSMILE-naive under heavy-tailed noise remains substantially less biased than SGHMC and SGLD were under Gaussian noise (see Table 1), confirming that the method does not rely on a strict Gaussian-noise assumption.

To quantify how the bias scales with the level of non-Gaussianity, and to probe the impact of skewness, we ran an experiment on the ICG target with centered log-normal noise,

$$\xi_i = e^{\sigma Z_i} - e^{\sigma^2/2}, \quad Z_i \sim \mathcal{N}(0, 1). \tag{25}$$

The noise is then rescaled via Equation (24) to have covariance $\mathbf{V}_{\text{corr}}$. The parameter $\sigma$ continuously controls the higher moments (skewness, kurtosis, ...) while leaving the first two moments of $\xi_i$ unchanged after rescaling.

As shown in Figure 4, for $\sigma < 0.1$ the bias is essentially flat, and the optimal step size is stable, implying pSMILE-naive is insensitive to mild non-Gaussianity. For $\sigma > 0.1$, higher-order moments start to dominate: the optimal step size shrinks to compensate, but the minimal achievable bias still grows under a fixed sampling budget, marking the regime in which non-Gaussianity becomes a first-order concern.

### D.3. Gaussian Mixture Models Experiments

To further validate the performance of pSMILE-naive on multi-modal distributions, we conducted a test on a 10-dimensional Gaussian Mixture Model (GMM).

**Setup.** The first two dimensions, $\theta_1, \theta_2$, of the parameter vector follow a 2D mixture of 25 Gaussian distributions:

$$p(\theta_1, \theta_2) = \frac{1}{25} \sum_{i=1}^{5} \sum_{j=1}^{5} \mathcal{N}\left((\mu_i, \mu_j), \Sigma_{\mathrm{GMM}}\right) \tag{26}$$

where the means are drawn from the grid set $\mathcal{M} = \{-4, -2, 0, 2, 4\}$ such that $\mu_i, \mu_j \in \mathcal{M}$. The covariance matrix for each component is set to $\Sigma_{\mathrm{GMM}} = 0.3\mathbf{I}_2$. The remaining eight dimensions follow a standard Gaussian distribution, $\theta_k \sim \mathcal{N}(0, 1)$ for $k \in \{3, \ldots, 10\}$. The ground truth density for the first two dimensions is visualized in the left panel of Figure 5.

**Implementation Details.** Following the setup in Section C, we inject isotropic random noise (the 'Isotropic' case). We sample the posterior using pSMILE-naive for $10^6$ steps with a fixed step size of 0.6. For comparison, we include cSGLD as a baseline, as it is known for its ability to explore multiple modes. We set a cosine schedule for cSGLD with 5,000 steps per cycle. Crucially, to ensure a fair comparison of exploration capability, we carefully tuned the step sizes such that the *average Euclidean norm of the update vector per step* was approximately equal for both methods.

**Results and Interpretation.** As shown in Figure 5, the single-chain pSMILE-naive (middle right) demonstrates a strong capability to traverse energy barriers and discover multiple modes, performing comparably to the cSGLD baseline (middle left) under equal update magnitudes. Furthermore, the multi-chain approach (right panel, 10 chains) successfully recovers all modes. This confirms that our method's exploration efficiency arises not just from the ensemble strategy, but also from the inherent dynamics of the sampler itself.

# E. Bayesian Neural Network Experiments

### E.1. UCI Benchmark

For the UCI benchmark presented in Table 2, we fit classical mean regression to the different tasks corresponding to the datasets described in Table 4. In the process, we always use a fully-connected feed-forward neural network with three hidden layers of size 16 each, resulting in about 700 total model parameters (depending on the input dimension). When sampling from the posterior, we use 1000 samples per ensemble member (chain), which is set to 10 members if not specified otherwise.

For the recently proposed Microcanonical Langevin Ensemble (MILE) method, we follow the setup of Sommer et al. (2025). Specifically, the DEs are optimized with the Adam optimizer with decoupled weight decay (Loshchilov & Hutter, 2019) and memberwise early stopping, after which sampling employs the auto-tuning strategy of MILE with 50k steps before providing 1k samples (following thinning of 10k samples). This leads to 60k full-batch sampling steps.

For all SGMCMC variants, we fix the step size of 0.001, which was individually verified to work best in terms of performance by a grid over both smaller and larger step sizes. For SMILE-naive, we even explored decaying step size schedulers, which did not improve the performance significantly. Furthermore, we both consider epoch-wise and batch-wise sampling. For the batch-wise case, we allow the stochastic variants to have the same computational budget as MILE in terms of sampling steps. As with the same number of mini-batch steps, the SGMCMC variants have not passed the data as often as MILE, so we also decided to compare with the epoch-wise sampling. This results in a number of batches times more sampling steps, as we only collect a sample after a full pass through the dataset. For hardware that can still handle full-batch updates well, this results in a stark computational overhead compared with the batch-wise and full-batch approaches, but might be considered a fairer comparison. For all SGMCMC experiments, we use a batch size of 256.

Further, each method in Table 2 is evaluated using three distinct train-test splits to assess the robustness of its performance.

For the batch and step size ablations summarized in Figure 6, we adopted the same setup as for Table 2, altering just the batch and step size of SMILE-naive and also considering one mini-batch and one full-batch configuration of scale-adapted SGHMC as baselines (for which we determined a suitable step size via grid search, and only the best performing step size 0.001 is reported).

### E.2. Image Classification (LeNet)

For these experiments, we adapt the CNN (v2) architecture from Sommer et al. (2025), which is a LeNet type of architecture (Lecun et al., 1998). We run an ensemble of 8 independent MCMC chains, each configured with 5,000 warmup steps followed by 10,000 sampling steps. Applying a thinning interval of 100 yields 100 samples per chain, for a total of 800

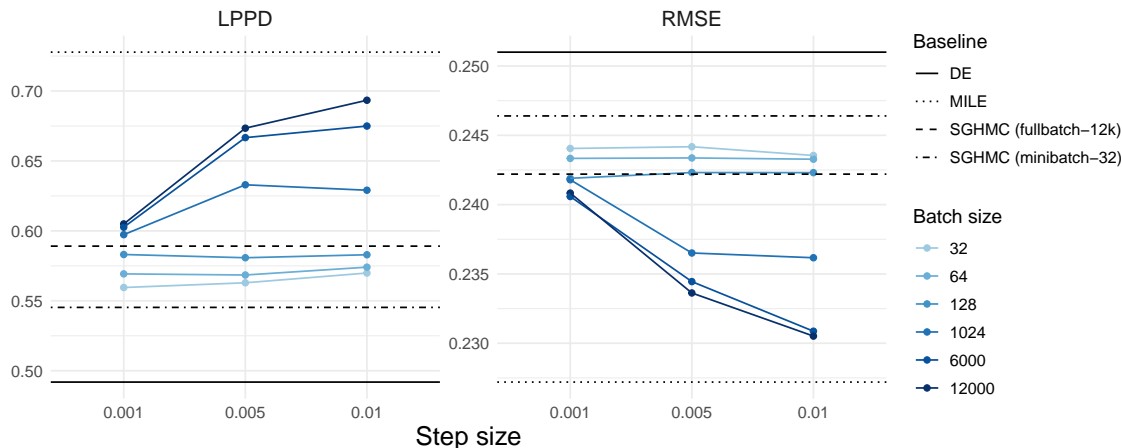

*Figure 6.* The performance of the SMILE-naive algorithm across various batch and step sizes in comparison with a few baselines on a distributional regression task for the `bikesharing` dataset. The SGHMC's step size is tuned, and the best performance is displayed.

final posterior samples used in the Bayesian model average and evaluation. Our analysis focuses on two key aspects: the empirically superior performance of pSMILE-naive against strong baselines and SMILE-naive (Table 6) and the robustness of SMILE to variations in mini-batch size (Figure 7). While larger batch sizes appear beneficial, the tested configurations suggest the improvements may exhibit diminishing returns. This suggests that SMILE can operate effectively even with moderate batch sizes, preserving much of its computational efficiency.

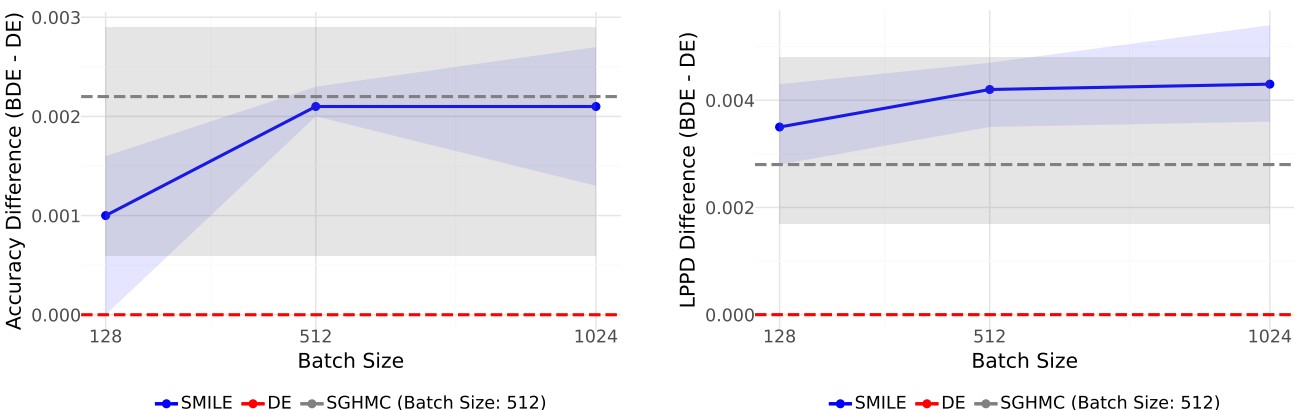

*Figure 7.* Relative performances with respect to a Deep Ensemble baseline of a Bayesian Deep Ensemble of LeNets (62k parameters) on the Fashion-MNIST dataset using different sampling routines. The shaded areas represent the minimal and maximal performance across three replications for the respective method. For both SGHMC and SMILE , we performed a grid search of suitable step sizes, and for both methods, 0.001 performed best.

*Table 6.* Relative performances with respect to a Deep Ensemble baseline of a Bayesian Deep Ensemble of LeNets (62k parameters) on the Fashion-MNIST dataset using different sampling routines. For all methods, we ablated over the step sizes of $[0.01, 0.001, 0.0001]$ and both for SGHMC and SMILE 0.001 performed best, for SMILE-naive 0.0001 and 0.01 for pSMILE-naive. The average performance across 3 replications is reported.

| | SGHMC | SMILE-naive | pSMILE-naive | SMILE |
|---|---|---|---|---|
| $\Delta$ Accuracy ($\uparrow$) | 0.0022 | $-0.0027$ | **0.0082** | 0.0021 |
| $\Delta$ LPPD ($\uparrow$) | 0.0028 | $-0.0362$ | **0.0101** | 0.0042 |

### E.3. Image Classification (ResNet-7)

The following details correspond to the results provided in Figure 1 and Figure 8. The architecture is a custom ResNet-7 with 428k parameters and Filter Response Normalization (FRN; Singh & Krishnan, 2020) instead of BatchNorm due to the critiques of BatchNorm in combination with sampling (Wenzel et al., 2020; Shen et al., 2024). Details on the architecture can be found in Table 7. We use an ensemble of 8 with 5k warmup steps, 10k sampling steps, thinning of 100, batch size 512 (if not indicated otherwise), standard normal isotropic priors, as well as various step sizes, and in the case of SMILE-naive, we also try out a cosine decay step size schedule.

*Table 7.* The Custom ResNet-7 Architecture with 428k trainable parameters. The output shape is specified for a sample input tensor of size $3 \times 32 \times 32$. All convolutional layers use a $3 \times 3$ kernel, stride 1, and 'SAME' padding, and are followed by Filter Response Normalization (FRN; Singh & Krishnan, 2020).

| Stage | Layer Operation(s) | Filters |
|---|---|---|
| input | Image | - |
| stem | Conv-FRN | 32 |
| body | Conv-FRN $\to$ MaxPool | 64 |
| | Conv-FRN | 64 |
| | Conv-FRN $\to$ MaxPool | 128 |
| | Conv-FRN $\to$ MaxPool | 128 |
| res_block | *(Identity Shortcut from previous output)* | 128 |
| | $\hookrightarrow$ Conv-FRN | 128 |
| | $\hookrightarrow$ Add | 128 |
| head | Global Average Pool | - |
| | Fully Connected | - |

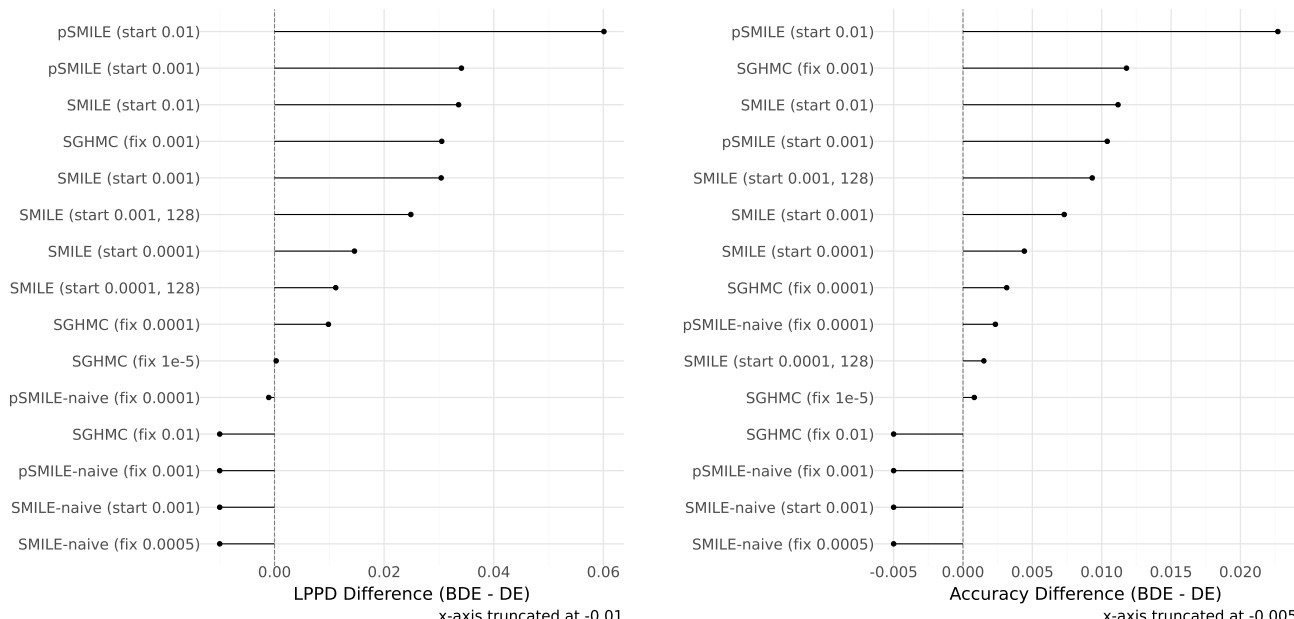

*Figure 8.* Relative performances difference between different Bayesian deep ensemble approaches and a deep ensemble baseline for a ResNet-7 (428k parameters) on the CIFAR10 dataset. The content of the brackets indicated whether a dynamic schedule with an initial step size or a constant step size schedule was used. If not indicated otherwise in the brackets, a batch size of 512 is employed. In each case, the performance of an ensemble of 8 chains is evaluated. Standard deviations over replications are comparable to those reported for the larger-scale setting reported in Table 9.

*Table 8.* The ResNet-18 Architecture using Filter Response Normalization (FRN; Singh & Krishnan, 2020) and 11.2M trainable parameters. The output shape is specified for a sample input tensor of size $3 \times 32 \times 32$. Each residual block (in square brackets) consists of two $3 \times 3$ convolutional layers. The first block of stages 2-4 uses a stride of 2 for downsampling.

| Stage | Layer Operation(s) | Filters |
|---|---|---|
| input | Image | - |
| stem | $3 \times 3$ Conv-FRN $\rightarrow$ MaxPool | 64 |
| stage1 | $\begin{bmatrix} 3 \times 3 \text{ Conv-FRN} \\ 3 \times 3 \text{ Conv-FRN} \end{bmatrix} \times 2$ | 64 |
| stage2 | $\begin{bmatrix} 3 \times 3 \text{ Conv-FRN, stride=2} \\ 3 \times 3 \text{ Conv-FRN} \end{bmatrix} \times 2$ | 128 |
| stage3 | $\begin{bmatrix} 3 \times 3 \text{ Conv-FRN, stride=2} \\ 3 \times 3 \text{ Conv-FRN} \end{bmatrix} \times 2$ | 256 |
| stage4 | $\begin{bmatrix} 3 \times 3 \text{ Conv-FRN, stride=2} \\ 3 \times 3 \text{ Conv-FRN} \end{bmatrix} \times 2$ | 512 |
| head | Global Average Pool | - |
|  | Fully Connected | - |

## E.4. Image Classification (ResNet-18)

The following details correspond to the results provided in Tables 3, 9, 10, 12 and 11. The architecture is a ResNet-18 with 11.2M parameters and Filter Response Normalization (FRN; Singh & Krishnan, 2020) instead of BatchNorm due to the critiques of BatchNorm in combination with sampling (Wenzel et al., 2020; Shen et al., 2024) as for the ResNet-7 model. Details on the architecture can be found in Table 8. We use an ensemble of 8 with 5k warmup steps, 10k sampling steps, thinning of 100, batch size 512, standard normal isotropic priors, as well as a step size of 0.001 and momentum decay 0.05 for scale-adapted SGHMC and a step size of 0.015 for SMILE. For cSGLD, we use a (peak) step size of 0.000001 and a target step size of 0.000000005 for four cycles and an exploration phase ratio of 33%, which is added to the sampling budget. All step sizes were determined by a shared grid over step sizes of 5-6 orders of magnitude.

*Table 9.* Image classification task on CIFAR-10 using a ResNet-18 with 11.2M parameters. Mean accuracy ($\uparrow$) and LPPD ($\uparrow$) results ($\pm$ standard deviation) are reported. All methods employ 8 ensemble members, and further details can be found in Section E.4.

| Method | LPPD ($\uparrow$) | Accuracy ($\uparrow$) | F1 ($\uparrow$) | Brier Score ($\downarrow$) | NLL ($\downarrow$) | AUROC ($\uparrow$) | AURC ($\downarrow$) |
|---|---|---|---|---|---|---|---|
| DE | $-0.3026 \pm 0.0056$ | $0.8995 \pm 0.0024$ | $0.8938 \pm 0.0046$ | $0.1545 \pm 0.0042$ | $0.4507 \pm 0.0018$ | $0.9934 \pm 0.0002$ | $0.0161 \pm 0.0007$ |
| cSGLD | $\mathbf{-0.2584 \pm 0.0048}$ | $\mathbf{0.9140 \pm 0.0018}$ | $\mathbf{0.9139 \pm 0.0019}$ | $\mathbf{0.1264 \pm 0.0024}$ | $0.4059 \pm 0.0031$ | $\mathbf{0.9954 \pm 0.0002}$ | $\mathbf{0.0111 \pm 0.0004}$ |
| SGHMC | $-0.2908 \pm 0.0021$ | $0.9104 \pm 0.0015$ | $0.9103 \pm 0.0015$ | $0.1389 \pm 0.0013$ | $0.4942 \pm 0.0060$ | $0.9949 \pm 0.0001$ | $0.0127 \pm 0.0002$ |
| SMILE | $-0.2763 \pm 0.0027$ | $0.9101 \pm 0.0019$ | $0.9100 \pm 0.0019$ | $0.1344 \pm 0.0014$ | $0.4071 \pm 0.0021$ | $0.9951 \pm 0.0001$ | $0.0122 \pm 0.0004$ |
| pSMILE | $-0.2598 \pm 0.0023$ | $\mathbf{0.9135 \pm 0.0009}$ | $\mathbf{0.9134 \pm 0.0009}$ | $\mathbf{0.1264 \pm 0.0009}$ | $\mathbf{0.3900 \pm 0.0045}$ | $\mathbf{0.9954 \pm 0.0001}$ | $0.0113 \pm 0.0002$ |

We further provide two optimization-based baselines in Table 10, namely IVON (Shen et al., 2024) and the Laplace approximation. For the Laplace approximation, the `posteriors` package (Duffield et al., 2025) is used. We run it for 300 epochs, a learning rate of 0.001, and a weight decay of 0.02. For IVON, we use the defaults of the accompanying code repository and suggestions of Shen et al. (2024) with the single Monte Carlo sample configuration.

## E.5. Image Classification (Vision Transformer)

We further extend our evaluation to a different architecture and dataset by training a Vision Transformer (Dosovitskiy et al., 2021) with patch size 16, hidden dimension 384, depth 12, 8 heads, and 22M parameters on Imagenette (Howard, 2019). For this experiment, we employed a batch size of 128 across 4 parallel chains. These results on this even larger scale BNN benchmark (Table 13) underscore the competitiveness of pSMILE . Based on these results, we can confirm that the gains of pSMILE cannot be attributed to step-size tuning or noise conditioning alone. Across both ResNet-18 and ViT settings, cSGLD and SGHMC benefit from the same tuning budget but do not match the performance level of pSMILE . The consistent improvements in all considered metrics indicate that the microcanonical formulation contributes materially and

*Table 10.* Image classification task on CIFAR-10 using a ResNet-18 with 11.2M parameters. Mean accuracy (↑) and LPPD (↑) results (± standard deviation) are reported. All methods are evaluated as single-mode approximations/chains, i.e., without ensembling across chains/mode approximations.

| Method | Accuracy (↑) | LPPD (↑) |
|---|---|---|
| DNN | $0.8541 \pm 0.0087$ | $-0.4394 \pm 0.0109$ |
| Laplace | $\mathbf{0.8915 \pm 0.0036}$ | $-0.4525 \pm 0.0345$ |
| IVON | $0.8735 \pm 0.0083$ | $-1.6163 \pm 0.0122$ |
| cSGLD | $\mathbf{0.8838 \pm 0.0038}$ | $\mathbf{-0.3649 \pm 0.0079}$ |
| SGHMC | $\mathbf{0.8717 \pm 0.0037}$ | $-0.3874 \pm 0.0090$ |
| SMILE | $\mathbf{0.8803 \pm 0.0038}$ | $\mathbf{-0.3541 \pm 0.0121}$ |
| pSMILE | $\mathbf{0.8858 \pm 0.0040}$ | $\mathbf{-0.3549 \pm 0.0184}$ |

*Table 11.* Ablation on reset quantile $\kappa$ of SMILE comparing predictive and UQ performance metrics for a single replication of the ResNet-18 experimental setup on CIFAR10 of Table 3.

| $\kappa$ | Accuracy | Brier Score | NLL | F1 Score | AUROC | AURC | LPPD |
|---|---|---|---|---|---|---|---|
| 0.90 | 0.9084 | 0.1328 | 0.3939 | 0.9082 | 0.9950 | 0.0123 | -0.2704 |
| 0.95 | **0.9127** | **0.1280** | **0.3931** | **0.9126** | **0.9954** | **0.0112** | **-0.2614** |
| 0.98 | 0.9108 | 0.1375 | 0.4159 | 0.9105 | 0.9948 | 0.0125 | -0.2863 |
| 0.99 | 0.9053 | 0.1444 | 0.4449 | 0.9051 | 0.9946 | 0.0132 | -0.3044 |
| No Guardrail | 0.0998 | 0.9384 | 12.3519 | 0.0182 | 0.5528 | 0.8828 | -2.5710 |

cannot be replicated by tuning or conditioning alone. Overall, these additions demonstrate that pSMILE constitutes a strong and robust performer relative to established adaptive SGMCMC methods.

### E.6. Language Modeling (NanoGPT)

The following details correspond to the results provided in Table 14, Figure 2, and the qualitative examples in Section F.6. The task is character-level language modeling on the `modern-shakespeare` dataset. The architecture is a 6-layer, 6-head GPT-style transformer with a context length of 256 and an embedding size of 384, with dropout disabled, adapted from Karpathy (2022). For model initialization, we employ a warmstart phase, training the model for 30 epochs using the Adam optimizer with decoupled weight decay with a batch size of 64 and a weight decay of 0.05. The learning rate is annealed linearly from $3e{-}4$ to $2.5e{-}4$. For the subsequent sampling phase, we use an ensemble of 4 chains with a batch size of 128. We run 200 warmup steps and collect 1000 posterior samples, thinned by a factor of 100. We use standard normal isotropic priors and a grid of the step sizes: $\{2e{-}6, 2e{-}5, 2e{-}4, 2e{-}3\}$.

### E.7. Ablations and Analysis of the Adaptive Tuning

**Tuning Evolution and Goodness of Fit** Figures 9 and 10 show that the adaptive step size and the parameters of the fitted Gamma distribution quickly converge to a stable regime. The system automatically finds and maintains a suitable level of energy error, which can differ by orders of magnitude between models. This highlights a key advantage: our method is more user-friendly in the context of BNNs than common MCLMC implementations that require manually setting a target energy error (Cabezas et al., 2024), as the user only needs to provide a reasonable initial step size like the optimizer's learning rate. The visualizations also confirm that the step size remains dynamic, biased toward decay for stability but capable of increasing to facilitate exploration. The effectiveness of this tuning is underpinned by the Gamma distribution's goodness of fit. With a target $\kappa$ of 0.02, the empirical reset frequency measured over tens of thousands of update steps for the NanoGPT model was $0.02314 \pm 0.00626$ and for the ResNet-18 model was $0.01778 \pm 0.00086$. This close alignment with the target confirms that the Gamma distribution provides a robust working model for the energy error.

**Hyperparameter Robustness Analyses** Both Table 12 and Table 11 confirm that the energy-variance-based tuning works robustly for a range of meaningful settings of the two core hyperparameters of Algorithm 1 $\kappa$ and adaptation probability $a$, with optimal performance achieved with moderate guardrailing and adaptation of the step size, even indicating that the strong performance of SMILE reported in Table 3 can be further improved. Notably, Table 11 clearly highlights the necessity to put the guardrail via $\kappa$ into place, as without this important component, the naive sampling completely diverges and

*Table 12.* Ablation on adaptation probability $a$ of SMILE comparing predictive and UQ performance metrics for a single replication of the ResNet-18 experimental setup on CIFAR10 of Table 3.

| $a$ | Accuracy | Brier Score | NLL | F1 Score | AUROC | AURC | LPPD |
|---|---|---|---|---|---|---|---|
| 0.00 | 0.9071 | 0.1475 | 0.4704 | 0.9069 | 0.9947 | 0.0130 | -0.3154 |
| 0.05 | 0.9065 | 0.1543 | 0.5144 | 0.9062 | 0.9945 | 0.0134 | -0.3379 |
| 0.10 | 0.9054 | 0.1391 | 0.4172 | 0.9053 | 0.9948 | 0.0128 | -0.2881 |
| 0.20 | **0.9096** | **0.1331** | **0.3879** | **0.9096** | **0.9950** | **0.0123** | **-0.2732** |
| 0.40 | 0.9046 | 0.1422 | 0.4060 | 0.9046 | 0.9944 | 0.0137 | -0.2941 |

*Table 13.* Ablation study of step sizes and schedules on Imagenette (ViT, 22M parameters). Results are reported for 4-member ensembles. Fixed (initial) step sizes are used for SGHMC and pSMILE, while cSGLD employs cyclical schedules. Runs of cSGLD with a higher peak step size of, e.g., $5 \cdot 10^{-6}$ and larger diverged catastrophically.

| Method | Step Size | LPPD ($\uparrow$) | Brier ($\downarrow$) | F1 ($\uparrow$) | Accuracy ($\uparrow$) | AUROC ($\uparrow$) | AURC ($\downarrow$) | NLL ($\downarrow$) |
|---|---|---|---|---|---|---|---|---|
| DE | - | $-0.9858$ | 0.3367 | 0.7702 | 0.7688 | 0.9691 | 0.0645 | 1.4957 |
| cSGLD | $10^{-6} \to 10^{-9}$ (4 cyc) | $-0.7815$ | 0.3325 | 0.7640 | 0.7638 | 0.9645 | 0.0691 | 3.4947 |
| cSGLD | $5 \cdot 10^{-7} \to 10^{-9}$ (4 cyc) | $-0.8122$ | 0.3300 | 0.7641 | 0.7641 | 0.9648 | 0.0683 | 2.6650 |
| cSGLD | $10^{-7} \to 10^{-9}$ (4 cyc) | $-0.9028$ | 0.3420 | 0.7606 | 0.7596 | 0.9651 | 0.0704 | 1.6526 |
| cSGLD | $5 \cdot 10^{-8} \to 10^{-9}$ (4 cyc) | $-0.9374$ | 0.3416 | 0.7619 | 0.7604 | 0.9659 | 0.0690 | 1.5850 |
| SGHMC | 0.00100 | $-1.0202$ | 0.4429 | 0.7058 | 0.7065 | 0.9475 | 0.1112 | $6 \cdot 10^{23}$ |
| SGHMC | 0.00050 | $-0.8769$ | 0.3874 | 0.7272 | 0.7268 | 0.9544 | 0.0927 | 2.2832 |
| SGHMC | 0.00015 | $-0.7841$ | 0.3440 | 0.7582 | 0.7578 | 0.9625 | 0.0758 | 1.2928 |
| SGHMC | 0.00010 | $-0.7893$ | 0.3419 | 0.7549 | 0.7544 | 0.9626 | 0.0755 | 1.2156 |
| SGHMC | 0.00005 | $-0.8239$ | 0.3458 | 0.7525 | 0.7518 | 0.9625 | 0.0762 | 1.2218 |
| pSMILE | 0.20 | $-1.0587$ | 0.4533 | 0.7030 | 0.7021 | 0.9535 | 0.0930 | 1.6831 |
| pSMILE | 0.15 | $-\mathbf{0.7708}$ | **0.3112** | **0.7738** | **0.7732** | **0.9712** | **0.0589** | **1.1985** |
| pSMILE | 0.10 | $-0.8929$ | 0.3247 | 0.7712 | 0.7708 | 0.9687 | 0.0628 | 1.3728 |
| pSMILE | 0.05 | $-0.9375$ | 0.3297 | 0.7693 | 0.7685 | 0.9692 | 0.0635 | 1.4352 |

performs poorly.

**A Practical Tuning Recipe** For practitioners seeking to deploy (p)SMILE, we recommend the following simple recipe:

1) Retain all hyperparameters except for the initial step size at their default values.
2) Set the initial step size close to the optimizer's final learning rate from the warm-start phase.
3) Evaluate a coarse step-size grid spanning 3 to 4 orders of magnitude. Empirically, even very short exploratory chains provide a reliable indication of stability and performance.
4) (Optional Refinement) Fine-tune the step size within the best-performing coarse region for a further performance boost.

Following this tuning, a rule of thumb that worked well in many tasks is to allocate approximately half of the initial optimization computational budget to the MCMC sampling phase. For a discussion on inference time considerations, see Sommer & Rügamer (2026).

**Parameter Interactions:** It is worth noting that the tuning parameters are not strictly independent. For instance, a larger initial step size naturally induces more extreme energy errors, thereby increasing the relevance of the numerical guardrail threshold ($\kappa$). However, as demonstrated by our ablation studies, the sampler maintains robust performance across a wide range of settings.

*Table 14.* Performance results for NanoGPT (10.8M) trained on `modern-Shakespeare` using the scale-adapted SGHMC, pSMILE and SMILE sampler.

| Sampler/ Step size | Accuracy (↑) | | | | Perplexity (↓) | | | |
|---|---|---|---|---|---|---|---|---|
| | DNN | BDE(1) | DE(4) | BDE(4) | DNN | BDE(1) | DE(4) | BDE(4) |
| **SGHMC** | | | | | | | | |
| 0.000002 | 0.5298 | 0.5337 | 0.5443 | **0.5495** | 5.0693 | 4.9094 | 4.4855 | **4.3710** |
| 0.000020 | 0.5290 | 0.5357 | 0.5451 | **0.5506** | 5.0547 | 4.8741 | 4.4503 | **4.3427** |
| 0.000200 | 0.5282 | 0.5441 | 0.5441 | **0.5563** | 5.0717 | 4.7170 | 4.4460 | **4.2601** |
| 0.002000 | 0.5302 | 0.2472 | **0.5470** | 0.1529 | 5.0227 | 22.0208 | **4.4194** | 10.9048 |
| **SMILE** | | | | | | | | |
| 0.000002 | 0.5289 | 0.5345 | 0.5428 | **0.5508** | 5.0803 | 4.9154 | 4.4955 | **4.3724** |
| 0.000020 | 0.5296 | 0.5381 | 0.5434 | **0.5526** | 5.0324 | 4.8393 | 4.4330 | **4.2980** |
| 0.000200 | 0.5296 | 0.5400 | 0.5458 | **0.5538** | 5.0779 | 4.7828 | 4.4663 | **4.3045** |
| 0.002000 | 0.5309 | 0.4900 | 0.5458 | **0.5527** | 5.0238 | 5.6973 | **4.4180** | 4.8501 |
| **pSMILE** | | | | | | | | |
| 0.000002 | 0.5290 | 0.5343 | 0.5464 | **0.5496** | 5.0678 | 4.8855 | 4.4744 | **4.3475** |
| 0.000020 | 0.5309 | 0.5376 | 0.5453 | **0.5531** | 5.0828 | 4.8699 | 4.4928 | **4.3540** |
| 0.000200 | 0.5290 | 0.5407 | 0.5457 | **0.5562** | 5.0374 | 4.7843 | 4.4507 | **4.3047** |
| 0.002000 | 0.5294 | 0.5318 | 0.5448 | **0.5530** | 5.0582 | 4.9221 | 4.4653 | **4.4454** |

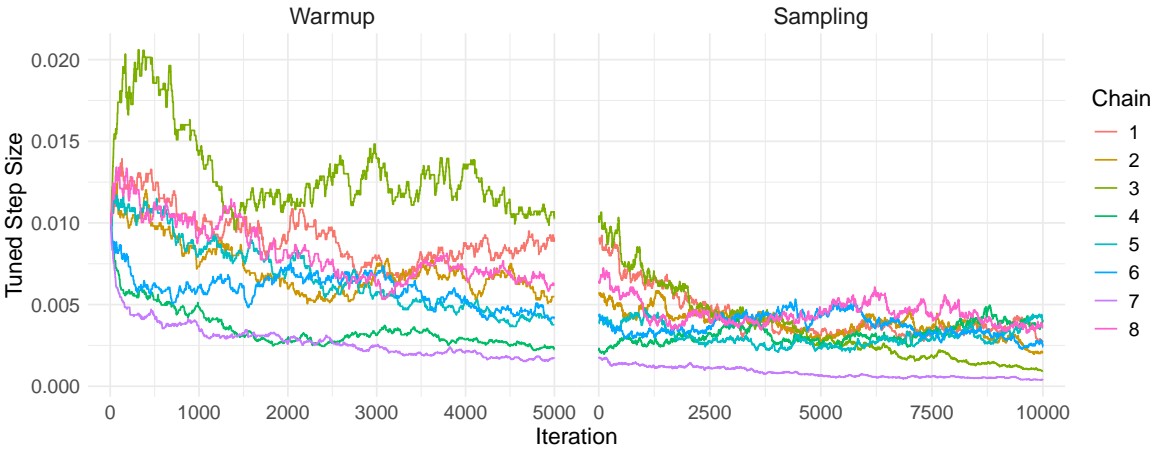

*(a)* Evolution of the step sizes over time for the SMILE sampler.

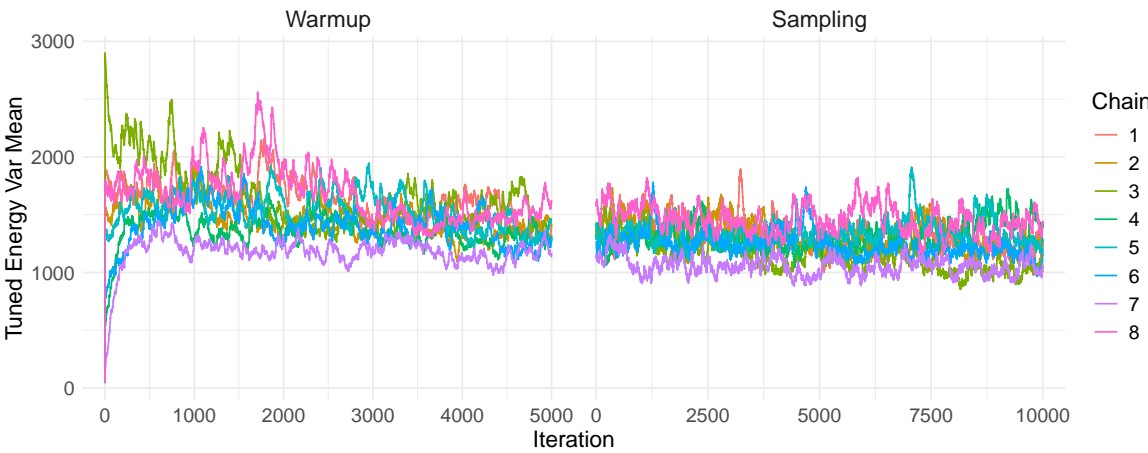

*(b)* Evolution of $\Delta E$ over time for the SMILE sampler.

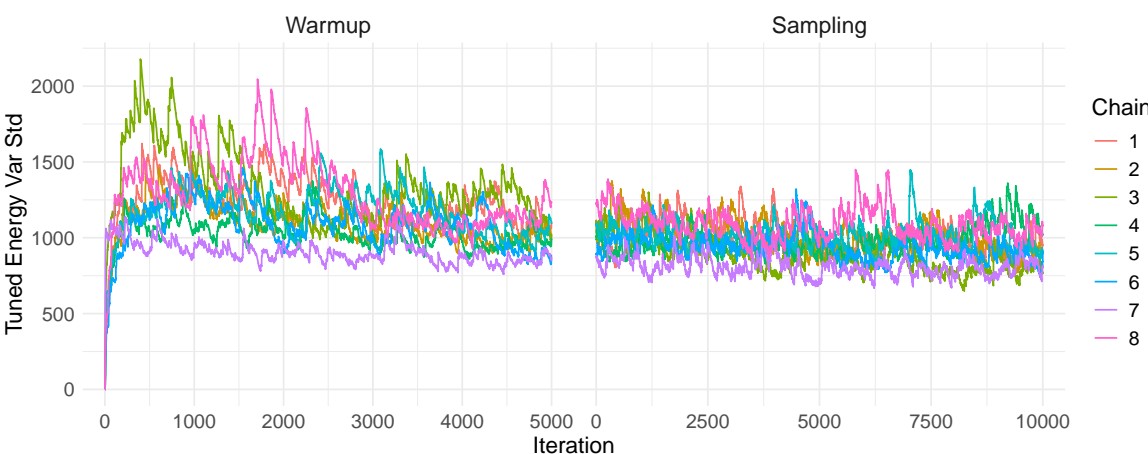

*(c)* Evolution of $SD(\Delta E)$ over time for the SMILE sampler.

*Figure 9.* Evolution of the key adapted quantities over time during the sampling of a ResNet-18 via SMILE , respectively, for 8 independent chains. The evolutions are depicted for the SMILE model in Table 3.

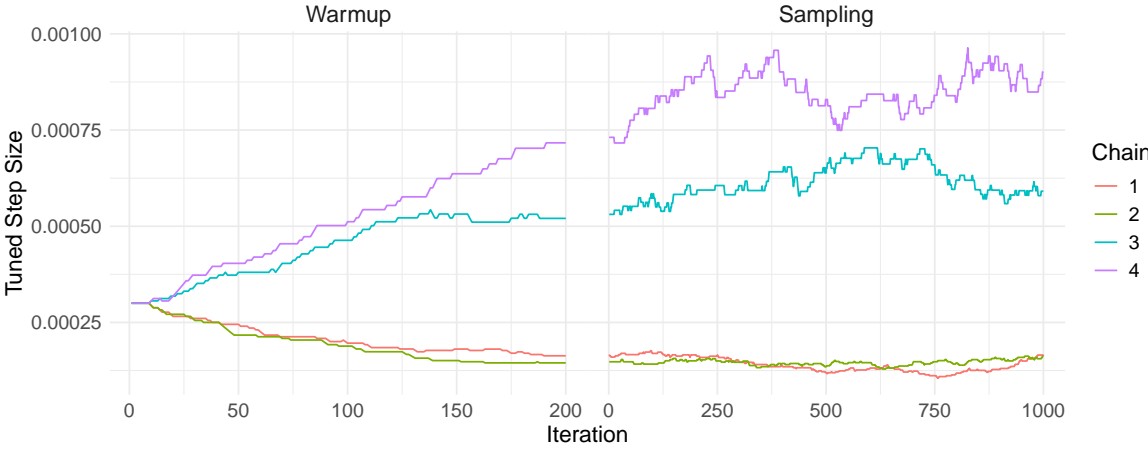

*(a)* Evolution of the step sizes over time for the SMILE sampler.

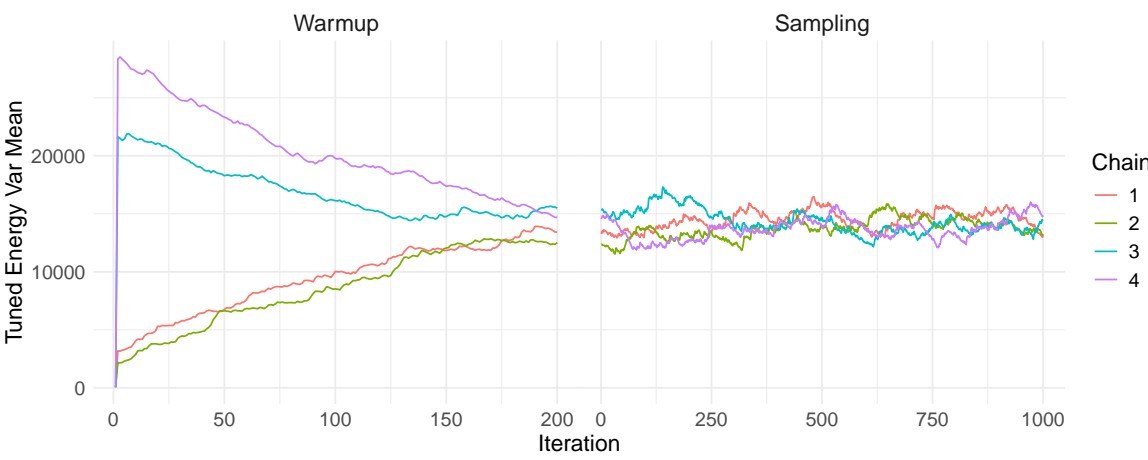

*(b)* Evolution of $\Delta E$ over time for the SMILE sampler.

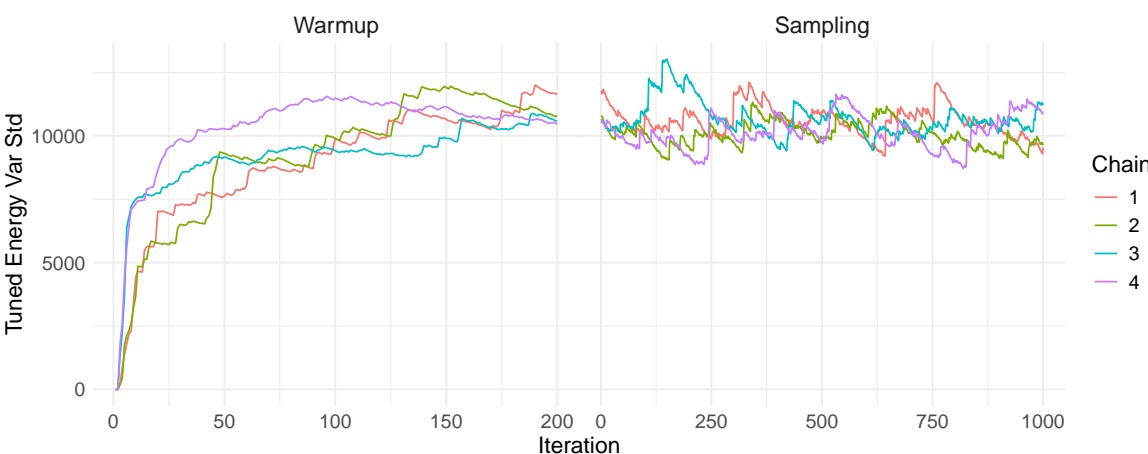

*(c)* Evolution of $SD(\Delta E)$ over time for the SMILE sampler.

*Figure 10.* Evolution of the key adapted quantities over time during the sampling of NanoGPT via SMILE , respectively, for 4 independent chains. The evolutions are depicted for the best performing SMILE model in Table 14.

# F. Supplementary Information

## F.1. Further Related Work on adaptive SGMCMC

Apart from scale-adapted SGHMC (Springenberg et al., 2016), various other adaptive SGMCMC approaches have been proposed in recent years. These include various stochastic gradient (Langevin) samplers which mimic pre-conditioned, momentum-driven, or adaptive optimizer behavior, such as Stochastic Gradient Fisher Scoring (Ahn et al., 2012), Precon-ditioned Stochastic Gradient Langevin Dynamics (Li et al., 2016), Momentum Stochastic Gradient Langevin Dynamics (Kim et al., 2022), or Cyclical SGLD (cSGLD; Zhang et al., 2020), methods that automatically tune the hyperparameters of SGMCMC, such as Multi-Armed MCMC Bandit Algorithm (Coullon et al., 2023), and, most recently, Parameter Extended SGMCMC (Kim et al., 2025) to improve SGMCMC's sample diversity and out-of-distribution performance.

## F.2. Conceptual Similarity with Cyclical SGLD

The resulting step size schedule of Algorithm 1 also shares conceptual similarities with cyclical SGLD (cSGLD Zhang et al., 2020): both introduce systematic variation of the step size to balance exploration and exploitation. However, our approach is stochastic rather than deterministic, producing diverse trajectories across chains, and decays only in expectation. This stochasticity enhances robustness and supports broader posterior exploration, while on average reproducing the empirically well-working decaying schedule of cSGLD in high-dimensional SGMCMC sampling. Further, as we rely on parallel ensembles, we effectively parallelize the sequential exploration and exploitation cycles of cSGLD and therefore increase effectiveness.

## F.3. Integrator Implementation Details

We implement the second-order minimal-norm Omelyan/McLachlan integrator (Omelyan et al., 2002; 2003), as used in the original MCLMC work. We use the standard 5-stage symmetric (P-Q-P-Q-P) scheme with coefficients: $b_1 = 0.1931833275037836$, $a_1 = 0.5$, $b_2 = 1 - 2b_1$.

This scheme splits the deterministic step into position updates (Q) and momentum updates (P). The position update (P) is a standard linear step, integrated for a time $\epsilon = a_1 \Delta t$:

$$\boldsymbol{\theta} \leftarrow \boldsymbol{\theta} + (a_1 \Delta t)\, \boldsymbol{\Pi}$$

For a momentum update of time $\epsilon$ (e.g., $b_1 \Delta t$ or $b_2 \Delta t$), the update $\boldsymbol{\Pi} \leftarrow B_\epsilon$ is given by:

$$B_\epsilon = \frac{\boldsymbol{\Pi} + (\sinh \delta + \boldsymbol{e}^\top \boldsymbol{\Pi}(\cosh \delta - 1))\boldsymbol{e}}{\cosh \delta + \boldsymbol{e}^\top \boldsymbol{\Pi} \sinh \delta}$$

where the auxiliary variables are defined using the gradient $g(\boldsymbol{\theta}) = \nabla \log p(\boldsymbol{\theta}|\mathcal{D})$ and dimensionality $d$:

- $\boldsymbol{e} = -g(\boldsymbol{\theta})/\|g(\boldsymbol{\theta})\|$
- $\delta = \epsilon \|g(\boldsymbol{\theta})\|/(d-1)$

A full deterministic step of size $\Delta t$ performs the following five operations in sequence:

1. P-step: $\boldsymbol{\Pi} \leftarrow B_\epsilon$ using $\epsilon = b_1 \Delta t$
2. Q-step: $\boldsymbol{\theta} \leftarrow \boldsymbol{\theta} + (a_1 \Delta t)\, \boldsymbol{\Pi}$
3. P-step: $\boldsymbol{\Pi} \leftarrow B_\epsilon$ using $\epsilon = b_2 \Delta t$
4. Q-step: $\boldsymbol{\theta} \leftarrow \boldsymbol{\theta} + (a_1 \Delta t)\, \boldsymbol{\Pi}$
5. P-step: $\boldsymbol{\Pi} \leftarrow B_\epsilon$ using $\epsilon = b_1 \Delta t$

Because MCLMC evolves the unit momentum direction $u = \boldsymbol{\Pi}/\|\boldsymbol{\Pi}\|$, this deterministic update is wrapped in a symmetric Strang splitting. Each full step begins and ends with a half stochastic/tangent update followed by normalization and reassignment.

This yields a reversible deterministic core with correctly projected stochastic evolution as shown in (Robnik et al., 2023).

### F.4. Gamma Quantile Approximation

We utilize the Wilson-Hilferty approximation for the efficient computation of the Gamma inverse cumulative distribution function.

Following Wilson & Hilferty (1931), given a probability $p$ and the standard normal quantile $z = \Phi^{-1}(p)$, the approximation is calculated as:

$$\mathcal{G}a^{-1}(p; \gamma_{\text{shape}}^{(t)}, \gamma_{\text{scale}}^{(t)}) = \gamma_{\text{scale}}^{(t)} \gamma_{\text{shape}}^{(t)} \left( 1 - \frac{1}{9\gamma_{\text{shape}}^{(t)}} + \frac{z}{3\sqrt{\gamma_{\text{shape}}^{(t)}}} \right)^3$$

### F.5. Runtimes

A brief note on computational cost. For the UCI benchmark (Table 2), the sampling runtime is on the same order of magnitude as the initial optimization time (with the only exception being the epoch-wise sampling, which is discussed above), a characteristic analyzed in Sommer et al. (2025). For our larger-scale experiments, we find that this practical feasibility holds. Sampling with the NanoGPT model took approximately 2.6 times longer than optimization. For the ResNet-18 on CIFAR-10, the sampling phase was only about 10% more costly than optimization. In both cases, the additional runtime invested into sampling is not excessive as in other works on sampling-based inference for BNNs (Izmailov et al., 2021), placing it in the same ballpark as optimization-based training while consistently yielding superior performance.

Further, in all considered BNN cases, the runtime of each step is dominated by the gradient calculation (via backpropagation). Therefore, given that the same integrator is used, there is no computational difference in practice. Obviously, SMILE requires less memory than pSMILE due to the preconditioning, but the same holds when comparing plain and preconditioned SGHMC.

## F.6. Qualitative Generation Examples

Below, we provide an example of a qualitative generation from one of the many fitted nanoGPT instances, demonstrating that it has learned meaningful syntactic and semantic structures despite the small vocabulary size (SingleChar Tokenizer).

**Model configuration:** `vocab_size = 65, context_len = 256, emb_size = 384, n_blocks = 6, n_heads = 6`

**Prompt I**: 'KIN'

**Generated Text**:
```
KING EDWARD IV:
No; that must I do well it mildle with of what
hors all my brother Angelo.  This grated is bourned
to the cowdich to save my heavinous, and lords,
whose I clim by your kindrow's doom!

DUKE OF AUMERLE:
As every me.  Duke of confidens bo my oweard.
I have smo not raitful and with the hour
```

**Prompt II**: 'Now is your time.  Speak loud and kneel before him.'

**Generated Text**:
```
Now is your time.  Speak loud and kneel before him.

DUCHESS OF YORK:
My lord come him.
Have your an mistivon!  What do you fremaid?

DUKE OF YORK:
Up the matter of the fight givive watch gentle Brotheren.
If you shall sham so my southern in or prevents;
some that find which coulding of the fast.

DUCHES OF YORK:
I have done the lords; and what often
```

*Figure 11.* Example texts generated by a trained nanoGPT instance (DNN) of Table 14 given the respective prompt and generating 300 new tokens autoregressively via sampling from the predicted Categorical distribution over the vocabulary. The model produces coherent theatrical formatting, consistent character names, and dialogue structure, indicating learned domain-specific patterns.

