# OpenReview forum: "Can Microcanonical Langevin Dynamics Leverage Mini-Batch Gradient Noise?"
_ICML.cc/2026/Conference — ICML 2026 regular_

### Official Review · Reviewer_qGLY · 2026-02-17

**Soundness:** 3
**Presentation:** 3
**Significance:** 2
**Originality:** 2
**Overall Recommendation:** 5
**Confidence:** 3

**Summary:**

This paper investigates the feasibility of scaling Microcanonical Langevin Monte Carlo (MCLMC) to large-scale Bayesian deep learning by replacing exact gradients with mini-batch stochastic gradients. The authors identify two primary barriers to this scaling: systematic bias caused by anisotropic mini-batch noise and numerical instability in high-dimensional parameter spaces.

To address these, the authors introduce SMILE (Stochastic Microcanonical Langevin Ensemble) and its preconditioned variant, pSMILE. The proposed framework incorporates a diagonal preconditioning scheme to mitigate noise-induced drift and a novel, energy-variance-based adaptive tuner. This tuner dynamically adjusts step sizes and establishes numerical guardrails by modeling the distribution of energy errors ($|\Delta E|$) with an online Gamma distribution. The method is evaluated across various tasks.

**Compliance With Llm Reviewing Policy:**

Affirmed.

**Final Justification:**

I appreciate the authors' detailed response. My concerns have been fully addressed, and I adjusted my score accordingly.

**Key Questions For Authors:**

- Algorithm 1 relies on several internal fixed parameters (e.g., $\delta = 0.02, a = 0.1, \beta = 0.01$). How were these specific values chosen, and is there a recommended "rule of thumb" for practitioners attempting to use SMILE on entirely new architectures?
- How sensitive is the performance of the adaptive tuner to these internal parameters across different tasks? For instance, do the optimal values for a Vision Transformer significantly differ from those for a NanoGPT model?
- Are the tuning parameters (like the adaptation probability $a$ and the reset quantile) entirely independent, or is there a correlation between them that could impact the stability of the Gamma distribution fit?

**Limitations:**

I think one critical theoretical limitation remains: convergence is formally guaranteed only when the mini-batch noise is isotropic. In practice, the proposed diagonal preconditioning only approximates this condition and does not ensure full isotropy. The authors already acknowledged that this approximation introduces a residual bias that might persist even in the limit of small step sizes.

**Strengths And Weaknesses:**

**Strengths**
- The paper is technically solid, with empirical evaluations spanning a wide range of modern architectures, such as ResNet-18 (11.2M parameters), Vision Transformers (22M parameters), and NanoGPT (10.8M parameters).
- The manuscript is exceptionally clear and well-structured. The use of "Pitfall" and "Remedy" boxes effectively highlights the core challenges and the authors' corresponding solutions.
- The work provides a clever adaptation of microcanonical dynamics (which are natively non-damped) to the stochastic regime without requiring explicit noise injection, a departure from standard SGMCMC conventions.
- By resolving the stability issues of MCLMC on large architectures, the authors provide a practical tool that allows the Bayesian community to leverage the high-dimensional efficiency of microcanonical samplers on modern datasets.

**Weaknesses**

While the authors provide a rigorous foundation for the continuous-time limit in the Appendix, there is a lack of convergence analysis for the proposed discretized stochastic algorithm ( finite-step behavior or an upper bound on the asymptotic bias introduced by the approximation).

---

> ### Author Rebuttal · Authors · 2026-03-30
>
> Dear Reviewer qGLY,
>
> Thank you for your positive feedback. We are grateful that you found the paper technically solid and appreciated both the range of architectures and the clear structure.
>
> **Convergence Analysis**
>
> > There is a lack of convergence analysis for the proposed discretized stochastic algorithm.
>
> We thank the reviewer for this comment and would like to emphasize three aspects we hope will change the reviewer’s assessment:
>
> 1. Our theoretical analysis in Appendix A already establishes both the preservation of geometric ergodicity and a precise characterization of the noise-induced drift, providing a solid continuous-time foundation.
> 2. Extending these results to a full finite-step-size convergence analysis is a non-trivial theoretical undertaking that we consider an important and natural next step. However, this challenge is shared across essentially all practical SGMCMC methods at the BNN scale.
> 3. Using a Taylor expansion of an Euler step (analogous to Sec 3.1 in Vollmer et al. 2016, JMLR), one can show that the noise introduces a bias of order $\mathcal{O}$(step size), and the first-order bias depends only on the covariance, not the full shape of the distribution. Preconditioning can reduce this first-order bias. Analysing the higher-order contributions and extending the local analysis to get a global bound remains an important and highly non-trivial future direction.
>
> We will discuss these points more explicitly in the revised manuscript.
>
> **Residual Preconditioning Bias**
>
> > the proposed diagonal preconditioning only approximates this condition and does not ensure full isotropy.
>
> We agree that this is an inherent limitation, but again shared by many related methods, such as scale-adapted SGHMC and pSGLD. As discussed in Section 5, the residual bias from the diagonal approximation is an open challenge wherever full covariance estimation is intractable. Our empirical results indicate that this approximation is sufficient to achieve state-of-the-art performance in practice, but we will make this limitation and its consequences more explicit.
>
> **Algorithm Configuration & Sensitivity**
>
> > Algorithm 1 relies on several internal fixed parameters [...] How were these specific values chosen [...]?
>
> We want to emphasize that we use a **single** fixed configuration for all internal hyperparameters across all BNN experiments, from ResNet-18 to Vision Transformers to NanoGPT. This, together with the ablations in Tables 10 and 11, demonstrates the robustness of our algorithm without requiring substantial expert knowledge to set internal parameters. The most influential remaining hyperparameter, as for all optimizers and samplers in deep learning, is the initial step size. Notably, pSMILE is substantially more robust to this choice than cSGLD and SGHMC (cf. Figure 2).
>
> To answer the reviewer’s question for a practical recipe: 1) keep all internal parameters at defaults, 2) set the initial step size near the optimizer's final learning rate, 3) run a coarse step size grid spanning about 3-4 orders of magnitude (in our experience, even short exploratory chains provide a reliable indication), 4) optionally refine the initial step size in the close vicinity to enable a further boost in performance. Then allocate roughly half the optimization budget for sampling.
>
> > How sensitive is the performance of the adaptive tuner to these internal parameters across different tasks?
>
> As outlined above, identical internal parameters transfer across all tested architectures without modification, confirming the generality of our defaults.
>
> **Parameter Independence**
>
> > Are the tuning parameters entirely independent, or is there a correlation?
>
> They are not entirely independent. For instance, a larger step size makes the guardrail more relevant as larger steps produce more extreme energy errors. However, the ablations show robust performance across settings, indicating no fragile interaction. We will add a brief discussion.
>
> ___
>
> While the focus of our paper is on the analysis and feasibility of stochastic microcanonical samplers, we believe the reviewer's suggestions regarding the practical recipe and parameter interactions will strengthen the paper. Please let us know if our response addresses your concerns and if there is anything else we can clarify. We thank the reviewer again for the constructive feedback!

---

> > ### Author Rebuttal · Reviewer_qGLY · 2026-04-03
> >
> > I appreciate the authors' detailed response. My concerns have been fully addressed, and I adjusted my score accordingly.

---

### Official Review · Reviewer_Likn · 2026-03-11

**Soundness:** 2
**Presentation:** 4
**Significance:** 3
**Originality:** 3
**Overall Recommendation:** 5
**Confidence:** 4

**Summary:**

Microcanonical Langevin Monte Carlo (mcLMC) is a recently introduced MCMC algorithm that has proven to be competitive or outperform HMC and other standard MCMC algorithms. This paper addresses the question of whether mcLMC can leverage stochastic gradients to accelerate sampling in large-scale applications. mcLMC is particularly attractive for use with stochastic gradients because the continuous-time version retains the correct stationary distribution no matter how much noise in injected, as long as it is constant and isotropic. The authors address a number of issues with a naive implementation of SG-mcLMC (they focus on an ensembled version they call SMILE). The key adjustments to ensure practical performance are an adaptive diagonal preconditioner to make the gradient noise constant, a "numerical guardrail" to avoid unstable updates, and a step size adaptation scheme to further improve stability in high-dimensional applications. Numerical experiments demonstrate that the proposed method pSMILE is much more robust to the choice of step size than alternative SG-MCMC algorithms, and as accurate or more accurate.

**Compliance With Llm Reviewing Policy:**

Affirmed.

**Final Justification:**

The authors have addressed my concerns

**Key Questions For Authors:**

None

**Limitations:**

Yes

**Strengths And Weaknesses:**

**Soundness:** The paper presents thorough experiments to support its claims and has some discussion of limitations. However, it is worth noting that -- like much of the SG-MCMC methodology in the ML literature -- it relies heavily on (seemingly reasonable) heuristics and lacks rigorous theoretical justification, beyond some characterizations of the limiting SDE.

**Presentation:** The paper is well written and generally easy to read.

**Significance:** The empirical results are sufficiently good that the proposed method has the potential to become a default option for SG-MCMC.

**Originality:** While the idea of creating a stochastic gradient version of mcLMC is quite natural, the authors execute it well. In particular, they do a very nice job of leveraging specific features of mcLMC such as its robustness to noise injection and the energy error. The modeling of the energy error and its use to adapt the step size in particularly creative.

---

> ### Author Rebuttal · Authors · 2026-03-30
>
> Dear Reviewer Likn,
>
> Thank you for the positive assessment and for recognizing that our method "has the potential to become a default option for SG-MCMC." We are particularly grateful for your appreciation of the creative leveraging of MCLMC-specific features.
>
> > It relies heavily on (seemingly reasonable) heuristics and lacks rigorous theoretical justification, beyond some characterizations of the limiting SDE.
>
> We want to clarify the scope of what is rigorously established in our paper. Our proofs (Appendix A) show that isotropic noise preserves stationarity and geometric ergodicity (Proposition 1), that anisotropic noise introduces bias (Theorem 3.1), and that correcting for noise anisotropy is theoretically necessary to approximately restore correctness, directly motivating our preconditioning scheme. These are formal results about the continuous-time dynamics, not heuristic arguments.
>
> The theoretical gap concerns the discretized algorithm with adaptive step sizes and guardrails. This is a challenge shared with virtually all practical SGMCMC methods at scale.  However, using a Taylor expansion of an Euler step (analogous to Sec 3.1 in Vollmer et al. 2016, JMLR), we can show that the noise introduces bias of order $\mathcal{O}$(step size), and the first-order error is only related to the covariance. Preconditioning can reduce this first-order bias, but analysing the higher-order contributions and extending the local analysis to get a global bound remains an important and highly non-trivial future direction. Since our method approximately corrects for the main sources of error, including discretization and noise anisotropy, we believe the residual error is practically small. In large-scale BNN settings, performance is primarily limited by Monte Carlo error from finite computational budgets rather than by algorithmic bias. Developing a complete finite-step-size theory for the minibatched microcanonical setting is an important yet non-trivial direction we plan to pursue.
>
> We also want to highlight that the practical components of our proposed approach are grounded in microcanonical-specific properties rather than being arbitrary. The energy error is a uniquely available diagnostic in MCLMC because the ideal dynamics conserve total energy, making it a natural basis for adaptation. The guardrails serve as a lightweight replacement for Metropolis-Hastings correction, which is prohibitive in high dimensions (Garriga-Alonso & Fortuin, 2021). The adaptive step size connects to classical adaptive MCMC theory (Andrieu & Moulines, 2006) but is specialized for energy-based diagnostics.
>
> ___
>
> We will further emphasize these points in a revised version of the paper and thank the reviewer for the helpful and encouraging feedback.

---

> > ### Author Rebuttal · Reviewer_Likn · 2026-04-02
> >
> > Thanks for your reply. I agree characterizing finite-step size behavior is challenging. You can cite https://arxiv.org/abs/2501.12212 as an example of this.

---

### Official Review · Reviewer_9kUx · 2026-03-13

**Soundness:** 3
**Presentation:** 3
**Significance:** 3
**Originality:** 3
**Overall Recommendation:** 5
**Confidence:** 4

**Summary:**

The paper asks whether it is possible to subsample gradients in the recently introduced microcanonical Langevin methodology for sampling from unnormalised densities. The answer is essentially "yes", provided that (i) anisotropy in the mini-batch gradient noise is accounted by appropriate estimation/conditioning, and (ii) numerical instabilities are taken care of through energy-error-based guardrails and adaptive step sizes.

**Compliance With Llm Reviewing Policy:**

Affirmed.

**Key Questions For Authors:**

Would it be possible to add an experiment in a controlled setting, showing how accurate the algorithm is in capturing the true posterior?

**Limitations:**

yes

**Strengths And Weaknesses:**

I think that this is a strong paper, which has a clear argument, supported by theory and practical methodology.

Soundness: The theory, methodology and experiments are sound and well developed (in particular the identification of anisotropic gradient noise is convincing). My main reservation: the final practical algorithm (including preconditioning, adaptive stepsizes, guardrails) appears to be biased (targeting an approximate posterior), and so it would be very helpful to have an experiment in a controlled setting to compare against the true posterior (and perhaps the influence of different design choices).

Presentation:  I find the presentation clear and easy to follow.

Significance: The paper addresses a key problem (scaling of Bayesian inference algorithms using mini-batch gradients).

Originality:  The findings are original and well thought through.

---

> ### Author Rebuttal · Authors · 2026-03-30
>
> Dear Reviewer 9kUx,
>
> Thank you for your positive assessment. We are grateful for your careful reading and suggestion for improvement.
>
> > The final practical algorithm (including preconditioning, adaptive stepsizes, guardrails) appears to be biased [...] it would be very helpful to have an experiment in a controlled setting to compare against the true posterior.
>
> In our controlled experiments (Table 1), we already compare against the true posterior using the exact squared bias metric. These experiments use pSMILE-naive (without guardrails), since at this scale, unbiased methods such as NUTS or the recently proposed adjusted MCLMC approach (Robnik et al., 2025) work without difficulty. The stability challenges that necessitate guardrails and adaptive tuning only emerge in high-dimensional settings.
>
> For large-scale BNNs, the true posterior is intractable. Importantly, all competitive SGMCMC methods in this regime carry approximation bias. The relevant practical question is whether the method achieves a favorable bias-variance trade-off under realistic budgets. Our results consistently show that pSMILE achieves superior or competitive performance across all metrics and architectures.
>
> To directly quantify whether we need the additional practical components to guard from catastrophic energy errors in the analytical setting of Table 1, we have analyzed the energy error distribution in those settings. We provide a new exemplary visualization in our anonymous repository: https://anonymous.4open.science/r/SMILE-icml-2026/data/energy_error_kde.png. As one can see, no catastrophic energy errors occur, so the numerical guardrails would not be triggered and are therefore not needed in these small-scale analytical posteriors.
>
> Further, we note that our existing ablations (Tables 10-11) already isolate the influence of individual design choices, such as the guardrail and adaptive step size, on the large-scale benchmarks, and we will connect these more prominently to the discussion of bias sources.
>
> ___
> We appreciate this constructive suggestion and believe it will further strengthen the paper. Thank you again for your positive evaluation! Please let us know if there is anything else we can clarify.

---

> > ### Author Rebuttal · Reviewer_9kUx · 2026-04-02
> >
> > Thank you for the clarification!

---

### Official Review · Reviewer_2ebU · 2026-03-15

**Soundness:** 2
**Presentation:** 3
**Significance:** 3
**Originality:** 4
**Overall Recommendation:** 5
**Confidence:** 4

**Summary:**

The paper evaluates a stochastic gradient extension of the microcanonical Langevin Monte Carlo (MCLMC) algorithm. Theoretically, MCLMC corresponds to the discretization of a stochastic differential equation involving a Brownian motion term. A peculiar property of  MCLMC is that it is invariant to the scaling of the "noise term." That is, the stationary distribution of the resulting Markov chain is the correct target distribution regardless of the noise scaling. Therefore, when using stochastic gradients in place of exact gradients, if the noise of the stochastic gradients followed an isotropic Gaussian distribution, MCLMC targets the correct stationary distribution. Now, even if the noise of the stochastic gradients did follow a Gaussian distribution, it would typically follow a position-dependent anisotropic Gaussian distribution. To this end, the submission considered preconditioning schemes to correct for this bias.

**Compliance With Llm Reviewing Policy:**

Affirmed.

**Final Justification:**

The authors have addressed my main concerns. That is, if the paper is aiming straight at BNNs, I believe the standards for analyzing correctness (of the stationary distribution) can be relaxed a bit. In light of this, I am in favor of acceptance. However, please do make this focus clearer in the next version.

**Key Questions For Authors:**

n/a

**Limitations:**

The paper does contain a limitation section. However, it could be made more detailed or clearer about what the proposed method is assuming, how those assumptions are failing, and the resulting consequences.

**Strengths And Weaknesses:**

## Strengths
* Developing better stochastic gradient Markov chain Monte Carlo (MCMC) is an important problem for scaling Bayesian inference to larger datasets.
* The way the submission leverages the fact that MCLMC is invariant to noise scaling for stochastic gradient MCMC is quite clever.

## Weaknesses
* The whole scheme relies on the crucial assumption that the stochastic gradient noise follows a Gaussian distribution. Previous work [1] stated that this is sensible if the central limit theorem holds. If this assumption is not satisfied, the proposed algorithm does not preserve the correct stationary distribution. Unfortunately, for minibatch gradients, the central limit theorem rarely holds. Or at least, the degree of violation must be theoretically or empirically evaluated. In particular, advances in the theoretical analysis of unadjusted MCMC algorithms have enabled us to analyze the asymptotic bias of stochastic gradient MCMC algorithms without relying on the Gaussian noise assumption. See Corollary 14 in [3] for example. Therefore, a theoretical analysis would very much strengthen the work.
* Related to the comment above, the empirical evaluations are inadequate for justifying the Gaussian assumption. The synthetic experiment in Table 1 satisfies the Gaussian assumption exactly and therefore does not provide insight into when it is violated. The larger-scale experiments in Table 2 involve Bayesian neural networks (BNNs), which only evaluate the predictive accuracy. Predictive accuracy is not a good metric for evaluating MCMC algorithms, especially unadjusted ones, since implicit bias could result in higher predictive accuracy at the expense of sampling accuracy. (This would be forgiven if this were a paper specialized to BNNs, but I believe this is not the case.)
* A key related work is missing. In particular, [4] argued that SGD can be viewed as a stochastic gradient Langevin algorithm if the stochastic gradient noise is an isotropic Gaussian. Since real-world stochastic gradients are rarely isotropic, they also propose to precondition the stochastic gradients to reduce bias in the stationary distribution.
* Minibatch MCMC algorithms are known to fail catastrophically when facing heavy-tailed/non-Gaussian target distributions (See the experiments in [5] with log-normal targets). It would be good to know whether MCLMC fares better on such problems.
* The blue boxes are not really doing much justice to the paper, and I recommend removing them. Specifically, the blue boxes are making claims that are too strong and require more evidence to be justified. Some of the evidence is provided in the experimental section, but not available to the reader at the point of reading the boxes.

## Additional Comments
* Line 15 "Hamiltonian Monte Carlo": HMC was first introduced in [6].
* Line 28 "MCLMC faces a critical limitation": This is not a limitation of just MCLMC but of virtually all MCMC algorithms.
* Line 40-41 "As the noise in stochastic gradient often resembles Langevin noise": Simply not true. At least, it is certainly not true for neural networks [7], but it is also rarely true for even linear regression.
* Eq (2): Cite the source of this equation.
* Line 120: Personally, I wouldn't cite Girolami and Calderhead (2011) for preconditioning in MCMC and certainly not for diagonal preconditioning. Preconditioning in MCMC goes back to [8] (Section 7.3) and adaptive MCMC [9].
* The way Section 2.3 is positioned feels a bit suboptimal to the reader. It would be much more natural to jump from 2.2 straight to 2.4.
* Line 215: "gradient anisotropy" -> "gradient **position-dependent** anisotropy"


1. Ma, Y. A., Chen, T., & Fox, E. (2015). A complete recipe for stochastic gradient MCMC. *NIPS*.
2. Zhu, Z., Wu, J., Yu, B., Wu, L., & Ma, J. (2019). The anisotropic noise in stochastic gradient descent: Its behavior of escaping from sharp minima and regularization effects. *ICML*
3. Durmus, A., Majewski, S., & Miasojedow, B. (2019). Analysis of Langevin Monte Carlo via convex optimization. *JMLR*.
4. Mandt, S., Hoffman, M. D., & Blei, D. M. (2017). Stochastic gradient descent as approximate Bayesian inference. *JMLR*.
5. Bardenet, R., Doucet, A., & Holmes, C. (2017). On Markov chain Monte Carlo methods for tall data. *JMLR*.
6. Duane, S., Kennedy, A. D., Pendleton, B. J., & Roweth, D. (1987). Hybrid Monte Carlo. *Physics Letters B*.
7. Simsekli, U., Sagun, L., & Gurbuzbalaban, M. (2019, May). A tail-index analysis of stochastic gradient noise in deep neural networks. *ICML*.
8. Roberts, G. O., & Rosenthal, J. S. (2001). Optimal scaling for various Metropolis-Hastings algorithms. *Statistical Science*.
9. Haario, H., Saksman, E., & Tamminen, J. (2001). An adaptive Metropolis algorithm. *Bernoulli*.

---

> ### Author Rebuttal · Authors · 2026-03-30
>
> Dear Reviewer 2ebU,
>
> Thank you for your detailed and constructive feedback. We are grateful for your recognition of the originality of our work and the care you put into your review.
>
> **Gaussian Noise Assumption**
>
> > The whole scheme relies on the crucial assumption that the stochastic gradient noise follows a Gaussian distribution. [...] for minibatch gradients, the central limit theorem rarely holds.
>
> We agree this deserves careful examination. The Gaussian assumption is used in our continuous-time analysis (Appendix A) to identify the noise-induced drift term. The key insight that anisotropic noise biases the stationary distribution is a structural property of microcanonical dynamics. While our formal derivations use the Gaussian framework, we expect this structural finding to extend beyond Gaussianity. Using a Taylor expansion of an Euler step (analogous to Sec 3.1 in Vollmer et al. 2016, JMLR), one can show that the noise introduces a bias of order $\mathcal{O}$(step size), and the first-order bias depends only on the covariance, not the full shape of the distribution. Preconditioning can reduce this first-order bias.
>
> Meanwhile, to directly evaluate the practical impact of violating this assumption, we extended Table 1 to include non-Gaussian gradient noise scenarios such as heavy-tailed distributions. Below we report results for pSMILE-naive under Gaussian, heavy-tailed Student-t, and Laplacian noise.
>
> | Target | Type | Gaussian | Student-t | Laplacian |
> | :-:|:-:|:-:|:-:|:-:|
> | ICG | Isotropic | 0.006±0.001 | 0.004±0.001 | 0.004±0.001 |
> | | Anisotropic | 0.038±0.008 | 0.071±0.020 | 0.072±0.020 |
> | | Correlated | 0.055±0.007 | 0.078±0.010 | 0.126±0.054 |
> | | Spatially-varied | 0.093±0.019 | 0.083±0.015 | 0.081±0.011 |
> | Rosenbrock | Isotropic | 0.004±0.001 | 0.007±0.001 | 0.009±0.001 |
> | | Anisotropic | 0.046±0.002 | 0.045±0.002 | 0.046±0.002 |
> | | Correlated | 0.070±0.005 | 0.063±0.028 | 0.064±0.009 |
> | | Spatially-varied | 0.052±0.005 | 0.068±0.019 | 0.049±0.009 |
> | Funnel | Isotropic | 0.021±0.005 | 0.016±0.004 | 0.022±0.002 |
> | | Anisotropic | 0.042±0.012 | 0.039±0.009 | 0.063±0.009 |
> | | Correlated | 0.004±0.002 | 0.036±0.009 | 0.019±0.006 |
> | | Spatially-varied | 0.012±0.003 | 0.084±0.027 | 0.061±0.019 |
>
> We observe comparable, sometimes slightly elevated bias under heavy-tailed noise. Crucially, pSMILE-naive remains substantially less biased than SGHMC and SGLD under Gaussian noise (cf. Tab 1), even in this non-Gaussian setting. Note that the benchmarks also feature highly non-Gaussian targets (Neal's Funnel, Rosenbrock).
>
> Finally, we want to highlight that our paper already provides rigorous theoretical contributions. We formally derive the noise-induced drift under anisotropic noise (Theorem 3.1, Appendix A.3), prove that isotropic noise preserves stationarity and geometric ergodicity (Proposition 1, Appendix A.2), and establish preconditioning as a theoretical necessity (if one does not want to follow the noise domination route, as also mentioned in 3.2 of our paper). We will discuss a more general non-asymptotic analysis as an important and non-trivial future direction and thank the reviewer for the critical assessment.
>
> **Evaluation Metrics**
>
> > The larger-scale experiments [...] only evaluate the predictive accuracy.
>
> We would like to clarify that our primary BNN metric is LPPD, which, in function space, measures how well posterior samples approximate the posterior predictive distribution (Gelman et al., 2014). LPPD is reported across all BNN experiments. The focus on function space metrics is rooted in the intractability of a ground truth and the non-identifiability of BNNs, rendering classical measures such as ESS and Gelman-Rubin statistic uninformative. We acknowledge the strong BNN focus and will state this more explicitly.
>
> **Related Work** Thank you for highlighting Mandt et al. (2017). We will incorporate this reference and discuss the conceptual connection: both works identify that anisotropic gradient noise necessitates preconditioning, though our approach operates in the microcanonical framework without explicit noise injection or damping.
>
> **Additional Comments** We thank the reviewer for the thorough list. We will add the HMC citation (Duane et al., 1987), revise Lines 28 and 40-41 citing Simsekli et al. (2019), incorporate the suggested preconditioning references (Roberts & Rosenthal, 2001; Haario et al., 2001), add the missing citation for Eq (2), and update Line 215. Regarding Section 2.3 positioning: could you please clarify? There is no Section 2.4 in our manuscript, so we want to make sure we understand the suggestion correctly (did you perhaps mean Section 4.4?).
>
> We believe the reviewer's suggestions, in particular the additional non-Gaussian noise experiments as well as the more detailed theoretical & related work discussion will notably improve the paper. Please let us know if there is anything else we can clarify.
>
> Thank you again for your constructive feedback!

---

> > ### Author Rebuttal · Reviewer_2ebU · 2026-04-03
> >
> > Thank you for the response.
> >
> > >  Using a Taylor expansion of an Euler step (analogous to Sec 3.1 in Vollmer et al. 2016, JMLR), one can show that the noise introduces a bias of order $\mathcal{O}$(step size), and the first-order bias depends only on the covariance, not the full shape of the distribution. Preconditioning can reduce this first-order bias.
> >
> > It is not obvious to me that this reasoning is very robust. Is the remainder obviously small even for heavy-tailed noise?
> >
> > > Meanwhile, to directly evaluate the practical impact of violating this assumption, we extended Table 1 to include non-Gaussian gradient noise scenarios such as heavy-tailed distributions. Below we report results for pSMILE-naive under Gaussian, heavy-tailed Student-t, and Laplacian noise.
> >
> > I appreciated the empirical evaluation. However, my concern has more to do with whether the effect of non-Gaussianity can be counteracted by controlling the step size. Furthermore, the evaluations don't tell us much except that the bias increases for non-Gaussian noise under anisotropy and correlations. Is this bad? How does the bias scale with the degree of non-Gaussianity?
> >
> > > Finally, we want to highlight that our paper already provides rigorous theoretical contributions. We formally derive the noise-induced drift under anisotropic noise (Theorem 3.1, Appendix A.3), prove that isotropic noise preserves stationarity and geometric ergodicity (Proposition 1, Appendix A.2), and establish preconditioning as a theoretical necessity (if one does not want to follow the noise domination route, as also mentioned in 3.2 of our paper).
> >
> > I don't quite see that Proposition 1 provides a rigorous theoretical understanding of the algorithm. The whole point of analyzing stochastic gradient methods is to analyze the effect of stochastic gradient noise under adverse conditions. This is precisely what is done in [1]. A formal analysis of what happens when the noise is *not* Gaussian would significantly strengthen the paper.
> >
> > > We would like to clarify that our primary BNN metric is LPPD, which, in function space, measures how well posterior samples approximate the posterior predictive distribution (Gelman et al., 2014). LPPD is reported across all BNN experiments. The focus on function space metrics is rooted in the intractability of a ground truth and the non-identifiability of BNNs, rendering classical measures such as ESS and Gelman-Rubin statistic uninformative. We acknowledge the strong BNN focus and will state this more explicitly.
> >
> > It is debatable whether LPPD truly measures function space approximation. A *bad* posterior approximation could still result in high LPPD as argued in [2]. However, since the Authors promised to more strongly state the focus on BNNs, I am willing to increase my score.
> >
> > 1. Durmus, A., Majewski, S., & Miasojedow, B. (2019). Analysis of Langevin Monte Carlo via convex optimization. JMLR.
> > 2. Deshpande, S., Ghosh, S., Nguyen, T. D., & Broderick, T. Are you using test log-likelihood correctly?. Transactions on Machine Learning Research.

---

> > > ### Author Response · Authors · 2026-04-04
> > >
> > > Dear Reviewer 2ebU,
> > >
> > > Thank you for your continued engagement and your willingness to raise the score. We address all remaining follow-up questions below.
> > >
> > > **Gaussian Noise Assumption.**
> > > > It is not obvious to me that this reasoning [Taylor expansion of an Euler step] is very robust. Is the remainder obviously small even for heavy-tailed noise?
> > >
> > > By expanding the discrete generator of the SMILE chain at finite step size $\epsilon$, assuming only noise expectation $\mathbb{E}[\xi]=0$ and noise covariance $\mathbb{E}[\xi^{}\xi^\top]= V(\theta)$, we can show that the leading noise-dependent error term is a diffusion operator on $S^{d-1}$ that depends only on the noise covariance $V$. When $V \propto I$, this is the Laplace-Beltrami operator, which preserves the stationary distribution exactly. Higher moments (skewness, kurtosis) enter only at $O(\epsilon^2)$ and beyond, so their contribution can be controlled by reducing the step size. We also verified the conclusion empirically with Student-t and Laplacian noise, as is shown in the previous table.
> > >
> > > > However, my concern has more to do with whether the effect of non-Gaussianity can be counteracted by controlling the step size.
> > >
> > > Yes, in the previous Student-t and Laplacian experiments, we ran a step size grid search optimized per noise type. At the optimal step size (which is smaller for heavier tails), the achievable bias is comparable or slightly higher than the Gaussian case, confirming that the effect of non-Gaussianity can be addressed by controlling the step size. We also reported the optimal step size in the lognormal noise case (see the new figure linked and described below), from which the shift towards smaller step size can be clearly seen when the non-Gaussianity increases. We thank you again for the comments, and will add this analysis and discussion to the revised manuscript.
> > >
> > > > Furthermore, the evaluations don't tell us much except that the bias increases for non-Gaussian noise under anisotropy and correlations. Is this bad? How does the bias scale with the degree of non-Gaussianity?
> > >
> > > To quantify scaling with the degree of non-Gaussianity, we conduct a new experiment using centered log-normal noise $\xi_i = e^{\sigma Z_i} − e^{\sigma^2/2} (Z_i∼N(0,1))$, rescaled to fixed variance. The parameter $\sigma$ continuously controls the higher moments while keeping the first two moments constant.
> > >
> > > The results are shown via the new figure https://anonymous.4open.science/r/SMILE-icml-2026/minimal_bias_vs_sigma.png. For $\sigma<0.1$, the bias remains constant and the optimal step size is stable. Beyond $\sigma \sim 0.2$, the higher order contribution becomes noticeable: the optimal bias results from using a smaller step size to compensate, but we still observe an increased bias under a fixed sampling budget.
> > >
> > > > *A formal analysis of what happens when the noise is not Gaussian would significantly strengthen the paper.*
> > >
> > > We agree that a full non-asymptotic analysis bounding the error under non-Gaussian noise, for instance along the lines of Durmus et al. (2019) [1], is a natural next step. As this is the first study of stochastic-gradient microcanonical dynamics, we aimed to cover both the theoretical fundamentals (continuous-time analysis, identification of failure modes, preconditioning as a remedy) and demonstrate practical viability through extensive and large-scale experiments. We believe we have done both. The discrete generator analysis and the new non-Gaussian experiments above already provide concrete steps toward characterizing the behavior beyond the Gaussian setting. We will discuss the full non-Gaussian analysis as an explicit future direction in the revised manuscript, in the context of our current empirical findings and with reference to [1].
> > >
> > >
> > > **Evaluation Metrics.**
> > > > It is debatable whether LPPD truly measures function space approximation. A bad posterior approximation could still result in high LPPD as argued in [2]. However, since the Authors promised to more strongly state the focus on BNNs, I am willing to increase my score.
> > >
> > > We absolutely agree with this nuance and thank the reviewer for the reference to Deshpande et al. BNN posteriors are inherently non-identifiable and singular, so no single metric perfectly captures posterior fidelity (neither in parameter nor in function space). We believe LPPD, together with other performance metrics, remains one of the best available evaluation approaches. We will cite [2], discuss this caveat, and, as noted earlier, more strongly state the BNN focus in the revised manuscript. We confirm this commitment.
> > >
> > > We are grateful for the constructive exchange throughout this review. In particular, the questions on non-Gaussian noise have pushed us to produce new interesting results that strengthen the paper materially.
> > >
> > > Thank you again!

---

### Decision · Program_Chairs · 2026-04-30

**Decision:**

Accept (regular)

**Comment:**

This paper considers the performance of microcanonical Langevin Monte Carlo with stochastic (subsampling) gradient noise. By default such noise would destroy the posterior distribution. However, if the sampling noise were Gaussian and known one could, essentially, reparameterize to render the noise Gaussian and isotropic, in which case the posterior would be preserved. This paper approximates the (non-Gaussian) noise as Gaussian, and then locally approximates it with a fully-factorized Gaussian. Empirically this seems to give reasonable results.